# Phytochemical Profiling, Isolation, and Pharmacological Applications of Bioactive Compounds from Insects of the Family Blattidae Together with Related Drug Development

**DOI:** 10.3390/molecules27248882

**Published:** 2022-12-14

**Authors:** Siwei Liang, Yifan Zhang, Jing Li, Shun Yao

**Affiliations:** 1School of Chemical Engineering, Sichuan University, Chengdu 610065, China; 2School of Life Sciences, Sichuan University, Chengdu 621065, China

**Keywords:** family Blattidae, phytochemical profiling, isolation, identification, bioactivities, drug development

## Abstract

In traditional Chinese medicine (TCM), insects from the family Blattidae have a long history of application, and their related active compounds have excellent pharmacological properties, making them a prominent concern with significant potential for medicinal and healthcare purposes. However, the medicinal potential of the family Blattidae has not been fully exploited, and many problems must be resolved urgently. Therefore, a comprehensive review of its chemical composition, pharmacological activities, current research status, and existing problems is necessary. In order to make the review clearer and more systematic, all the contents were independently elaborated and summarized in a certain sequence. Each part started with introducing the current situation or a framework and then was illustrated with concrete examples. Several pertinent conclusions and outlooks were provided after discussing relevant key issues that emerged in each section. This review focuses on analyzing the current studies and utilization of medicinal insects in the family Blattidae, which is expected to provide meaningful and valuable relevant information for researchers, thereby promoting further exploration and development of lead compounds or bioactive fractions for new drugs from the insects.

## 1. Introduction

Over the centuries, natural products, especially metabolites of plants, have become a valuable source of medicine for human beings. Nevertheless, there has been a lack of exploration and development of the medicinal substances from animals, particularly insects. Even though their number and variety are significantly less than those of herbs, medicinal insects are an essential part of traditional ethnomedicine, and their bodies, secretions, derivatives, and pathological products can be used to treat a variety of ailments. In general, their bioactive constituents exhibit stronger pharmacological effects than those of plants. As the famous representative, the insects of the family Blattidae have a long history of medicinal application, and some studies have demonstrated their antitumor [1,2,3], tissue repair [4,5,6], antibacterial [7], antiviral [8], and other pharmacological effects [9,10]. The family Blattidae belongs to Arthropoda, Insecta, Pterygota, Blattariae in entomological taxonomy, which has existed on the earth for 320 million years [11]. The species is extremely widely distributed throughout the world and can be found almost anywhere except the poles due to its high reproductive potential and adaptability. This family is well known for its role as a sanitary pest throughout the world [12].

On the other hand, the family Blattidae can still survive and reproduce when facing thousands of powerful insecticides, which indicates that they have extremely strong vitality and should contain certain special substances in their bodies with special functions [13]. Meanwhile, the number of insects in this family is excellent, which indicates that there must be considerably effective ways to defend themselves against various possible infections. In 1960, it was reported that there were about 3500 species of Blattidae in the world. Over the years, many new species have been discovered. According to “Biological Abstracts” and “Zoological Record”, 5147 species of Blattidae have been recorded worldwide as of five years ago, of which 253 were recorded in China [14]. As reflected by the quantity, this is clearly a great potential medicinal resource. Most of Blattidae lives in the wild, and a few species live indoors. Among them, the species closely related to human beings are mainly *Periplaneta americana* (*PA*, American cockroach), *Periplaneta australasiae*, *Periplaneta fuliginosa*, *Blattella germanica*, and *Blatta orientalis*, which comprise the main medicinal insects of Blattidae. In particular, the American cockroach is the most studied object, with the largest body size in this family [15], which has been artificially cultivated to meet the large-scale needs of the pharmaceutical industry.

In China, the family Blattidae has been applied as traditional Chinese medicine for the treatment of wounds, burns, ulcers, and primary liver cancer, which was first recorded in the medicinal book “Shennong’s Classic of Meteria Medica” in ancient times [16]. In the early stage, the fresh adults of the Blattidae were used immediately after execution or could be scalded with boiling water and then dried for medicinal use (in dosage forms like decoctions, pills, or powder preparations; internal or external use). Currently, processing and preparation technology is becoming increasingly complex and scientific. In addition to extracts, defatted refined powders of Blattidae, which no longer have unpleasant appearance, properties, or flavor, can also be used as health products with the effect of promoting blood circulation or dredging tendons and muscles [17]. In recent years, research on the family Blattidae has focused on: (1) the relationship of growth, environment, physiology, and biological characteristics; (2) whole-genome sequencing, analysis, and gene expression; (3) chemical constituents and pharmacological effects; (4) preparation process of related drugs and DNA vaccines, etc. Currently, a large number of previous studies on the chemical composition of Blattidae demonstrated that the bioactive components are divided into the following categories: proteins (including enzymes), amino acids, and peptides; polysaccharides; fatty acids; pheromones; and polyols, alkaloids, and organic acids [18]. There is a summary of the number of works on different chemical components in the family Blattidae in past years (see Figure 1). Although various patented drugs and preparations made of Blattidae have shown good pharmacological activities, the key chemical components with active functions and their structures still have not been clearly discovered. Additionally, the quality assessment of Blattidae, preparations, and extracts still focuses on the qualitative detection and quantitative analysis of a few index components. However, the fact should be that several active fractions play a synergistic role based on a reasonable proportioning relationship rather than acting alone. Therefore, it is difficult to control the effectiveness and safety of insect medicines effectively with a limited number of ingredients as indicators. Therefore, it is essential to further explore the related basic issues comprehensively to promote the development and clinic application of the family Blattidae. On the basis of the above background, this review aimed to summarize the phytochemical profiling, isolation, and pharmacological effects of bioactive substances from insects of the family Blattidae as well as the current situation of related drug development. In this way, sufficient attention would be paid to the discussion and analysis of existing problems. This is expected to provide a meaningful reference to researchers focusing on medicinal insects.

## 2. Phytochemical Profiling

For a clearer distinction, the chemical components in the family Blattidae include two major sources: those released from the worm and those present in the bodies of the insects. Based on the methods of preparing Blattidae as medicine, the latter is generally regarded as the main basis of effective substances. The chemical components released by the worms are composed of sex pheromones, mainly including sesquiterpenes and aggregation pheromones; they are mostly small-molecule organic acids, ammonia, and hydrocarbons. The chemical components in the insect body attract more attention due to their higher contents and are mainly composed of small molecules (amino acids, polyols, isocoumarins, alkaloids, flavonoids, nucleosides, oligopeptides, etc.) and macromolecules (proteins, polysaccharides, peptides, etc.) [19]. In addition, there are lipids, terpenes, and steroids found in the family Blattidae.

### 2.1. Amino Acids, Peptides, and Proteins

Among the chemical components present in the bodies of Blattidae insects, various amino acids, small peptides, and proteins are found in abundance. As the most important constituents, these three types of molecules in Blattidae have shown high nutritional and medicinal values. Moreover, in the body of *PA*, the total amount of amino acids (AAs) is high, with a mass fraction of up to 43.14%. There are seventeen types of amino acids, which are shown in Table 1, including seven types of essential AAs (threonine, valine, methionine, isoleucine, leucine, phenylalanine, lysine) as well as drug-effective AAs. The content of the former accounts for 35.37% of the total AAs, and the content of medicinal amino acids, such as glutamic acid and aspartic acid, reaches up to 55.43% of the total AAs [20,21]. According to the ideal model of FAO/WHO, the amino acid composition of better-quality protein is nearly 40% essential AAs/total AAs, indicating that the *PA* is a protein resource with high nutritional value.

Of the peptides mainly contained in Blattidae, one type is neuropeptides, which are a class of small molecular peptides that regulate a series of physiological processes, such as insect growth and development, molting metamorphosis, metabolism, and reproduction. Up to now, more than 50 neuropeptides have been discovered, including allatostatin, pyrokinin, kinin, and periviscerokinins [22]. The other type is antibacterial peptides, which have good thermal stability and acid resistance. A series of cyclic dipeptide compounds was isolated from the ethyl acetate extraction site of *PA*, and these compounds were identified as 8-hydroxy-3, 4-dihydroquinoline-2 (1H)-one, cyclo-(L-Phe-L-Pro), cyclo-(Pro-Ile), cyclo-(L-Pro-D-Leu), brevianamide F, cyclo-(Ile-Ala), cyclo-(L-Val-L-Pro), cyclo-(L-Pro-L-Tyr), cyclo-(tryptophan-valine)-Dipeptide, and (-)-(1S, 3S)-1-methyl-1,2,3,4-tetrahydro-β-carboline-3-carboxylic acid [23]. The research conducted by Yang et al. shows that the main pharmacodynamic substance of “Kangfuxin” liquid made of *PA* extract could be small active peptides, with molecular weight below 15 kD, which are composed of the aforementioned amino acids [24]. In another case, the ethanolic extract of *PA* was loaded on HP-20 macroporous resin and eluted with 70% ethanol, and then the target fraction was obtained after lyophilization. As a result, tricine-SDS-PAGE gel electrophoresis was used to confirm that the fraction with antitumor activity was a peptide mixture with a relative molecular weight around 3 kD [25]. Wang’s group obtained 14 cyclic dipeptides from *PA* via chromatographic separation, and identified them as cyclic-(proline-threonine), cyclic-(proline-alanine), cyclic-(alanine-valine), cyclic-(glycine-isoleucine), cyclic-(glycine-leucine), cyclic-(leucine-serine), cyclic-(serine-phenylalanine), cyclic-(asparagine-phenylalanine), cyclic-(tyrosine-glycine), cyclic-(tyrosine-alanine), cyclic-(tyrosine-serine), cyclic-(tyrosine-threonine), cyclic-(tyrosine-aspartate) using spectral analysis [26]. Although the research on antibacterial peptides of the family Blattidae is in the preliminary stage, as far as the existing research reports are concerned, antibacterial peptides have been found with strong antibacterial activity, which should be paid more attention. Finally, there are two types of peptides found in their corpus cardiacum—hyperglycemic hormones I and II—which provide a new direction for exploring peptides with other meaningful activities.

Finally, insect protein is the third largest source of protein after microorganisms and cellular organisms. Due to the large number of types, fast reproduction, high protein content, low fat and cholesterol content, and easy absorption by the human body, insect protein is a source of high-quality macromolecules and has attracted the interest of many researchers around the world. In particular, the protein content of Blattidae is higher than that in other fish and livestock. According to the analysis for *PA*, the crude protein contents of adults and nymphs were 61.9%, 67.2%, and 60.2%, respectively. Yao et al. conducted SDS-PAGE analysis of the proteins of three cockroaches (*Periplaneta fuligionsa, PA, Blattella germanica*) and found that the number and location of related protein bands were different [27]. The numbers of male and female protein bands in *Periplaneta fuligionsa*, *PA*, and *Blattellager manica* were 29, 31; 28, 30; and 25, 28, respectively. Moreover, the number and location of protein bands of *Periplaneta fuligionsa* and *PA* between 25.7 kD and 22.4 kD were similar, and there were differences around 100 kD. In addition, there were also differences between males and females, mainly around 61.7 kD; females had protein bands but males did not. Furthermore, the number of protein bands of *Blattella germanica* were clearly less than the former two. For example, there was an extra protein band at 2.09 kD, while such a band was not found in the former two.

In summary, the research focuses on active peptides of *PA*, and the focus on such antibacterial compounds will be the main research direction in the future. Related antimicrobial peptides can act on the bacterial cell membranes and kill bacteria by neutralizing the electric charge, which greatly reduces the possibility of bacterial resistance and makes them a potential drug to replace traditional antibiotics. Even better, the current research foundation is also increasingly stable. On the one hand, the types and contents of amino acids have been thoroughly studied as structural units; on the other hand, antimicrobial peptides with many potential activities are being discovered [28,29,30,31], especially in the aspect of antitumor effect and immunity promotion.

### 2.2. Polysaccharides

A polysaccharide is a polyhydroxy macromolecule composed of 10 or more monosaccharides linked by glycosidic bonds [32]. It is a type of essential biological polymer in the natural world (especially in medicinal insects), juxtaposed with proteins and nucleic acids, which plays an essential role in different life activities. Generally, polysaccharides have extremely complex and diverse structures and properties, so they have a variety of important physiological functions—such as antitumor and immune regulation [33]—which has attracted increasing attention in the fields of medicinal chemistry and pharmaceutics. Animal polysaccharides mainly include chitin, chitosan [34], glycogen, and polysaccharide complexes such as proteoglycans, lipopolysaccharides, etc. A large amount of glucosamine compounds has been found in the family Blattidae. Most active medicinal insect polysaccharides are β-(1-3)-D-glucans with β-(1-6) single glucose residue side chains, and β-(1-6) glycosidic bonds on the glucose backbone are necessary for antitumor activity such as in compound 1 [35] (see Figure 2). As the known sixth important factor, the content of chitin in dried shells of Blattidae reaches 12% to 15% [36]. The shells remaining after extraction can also be used to produce chitin, chitosan [37], and their derivatives, and it is very beneficial to realize a comprehensive utilization for this insect [38]. In addition, the active insect protein is rich in chitin, and the purity of the chitin contained in the insect protein is several times greater than in shrimp and crab, which makes its utilization more meaningful [34,39].

Additionally, this type of macromolecule is also found inside the body of Blattidae. Measurement of the total sugar of *PA* in different breeding bases via the phenol–sulfuric acid method showed that the total sugar content of the insects in different bases was 15.56–45.65 mg/g, which was quite different between various individuals; this may be affected by related factors, such as feed, breeding cycle, collection, and processing during cultivation [40]. Sometimes, the complex between them and other types of compounds is more interesting to researchers. The same research has also shown that the sticky sugar amino acid (SSAA) is the main active component of the American cockroach for the clinical treatment of viral hepatitis. SSAA is a type of macromolecular polymer with a molecular weight range of 1–8 kD (average 5 kD) in a nearly normal distribution. After hydrolysis with hydrochloric acid, it was determined that it was mainly composed of amino acids, sulfate, glucose, glucuronic acid, and glucosamine. Heparan sulfate (HS) is the main component of sulfated glycosaminoglycans isolated from the family Blattidae, accounting for more than 90%. In the body fat and pectoral muscle of *PA*, the content of HS is very dominant. Santos et al. found that the highest content of SSAA in adult males is HS accounted for more than 90% of the total content, which can be detected in the thymus, muscle, digestive tract, and reproductive system [41]. There are several studies on the isolation and purification of crude polysaccharides from *PA* to obtain a group of neutral polysaccharide components and a glycoprotein complex with a sugar content of 458 g/kg and a protein content of 392 g/kg. The composition was analyzed and found to include xylose, arabinose, glucose, mannose, galactose, glucuronic acid, and galacturonic acid [42].

### 2.3. Nucleosides

Nucleosides are often used as primary antiviral and antitumor therapeutic drugs. The natural nucleosides present in animals and plants can generally be divided into pyrimidine nucleosides and purine nucleosides [43]. The current research on nucleosides is mainly focused on structural modification of the base and glycosyl to obtain drugs with different activities. Nucleosides, as the basic components of living organisms [44], are also one of the main body chemical components of the family Blattidae. The main nucleoside components are reported as inosine (**2**), hypoxanthine (**3**), and uracil (**4**), which are shown in Figure 3. In the current report, the contents of total nucleic acid in the *PA* powders before and after fermentation were found to be 33.80 mg/g and 22.83 mg/g, respectively [45]. Wu et al. identified hypoxanthine, trigonelline, uracil, adenosine, and inosine from the extract and preparations [46]. Studies have shown that the main active ingredient of the traditional Chinese medicine preparation “Xinmailong Injection”, refined from the extract of *PA*, could be composed of nucleoside compounds.

### 2.4. Polyols

Polyols can be used to treat viral myocarditis, explosive deafness, and epidemic encephalitis B. Chen et al. determined the polyols contained in the ethanol extract of American cockroaches, including fructose, mannitol, and inositol [47], and the content of mannitol was 1.8% of the defatted product in American cockroaches. It was also confirmed that polyols accounted for the main proportion of the volatile components of the American cockroach, identified as glycerin and 2,3-butanediol [48]. The former is known as a skin-replenishing ingredient which is a cornerstone of many moisturizers, and the latter is also an important microbial metabolite composed of several stereoisomers. In general, there are quite few studies on polyol components in Blattiade. In many cases, they are regarded as the subordinate components related to the pharmacological effects of the family, so they do not attract enough attention. Although the structure is in actuality simpler and less novel than the aforementioned components, the synergistic action provided by them should not be ignored.

### 2.5. Isocoumarins

Lower concentrations of isocoumarins than coumarins exist in herbs, and the former are more easily found in animals, fungi, bacteria, and even marine micro-organisms. Firstly, the 70% alcohol extract of *PA* was separated and seven types of isocoumarins were obtained (see Figure 4). Among them, (3R)-methyl-6,8-dihydroxy-7-formoxyl-3,4-dihydroisocoumarin (**5**), (3R)-methyl-6,8-dihydroxy-7-methyl-3,4-dihydroisocoumarin (**6**), (3R)-ethyl-6,8-dihydroxy-3,4-dihydroisocoumarin (**7**), (3S)-hydroxymethyl-6,8-dihydroxy-7-methyl-3,4-dihydroisocoumarin (**8**) are new compounds and (3R)-ethyl-6,8-dihydroxy-7-methyl-3,4-dihydroisocoumarin (**9**), (R)-6-hydroxymellein (**10**), and (3R)-methyl-7-hydroxymethyl-8-hydroxy-3,4-dihydroisocoumarin-6-*O*-β-D-glucopyranoside (**11**) are known compounds [49]. When the above components were tested for cytotoxic activity using the MTT method, it was found that compounds **7** and **8** were new compounds and that compound **9** had the effect of inhibiting the activity of tumor cells, i.e., Hep G2 and MCF-7 cells. Xu et al. also discovered pericanaside (**11**), which is an unreported compound of dihydroisocoumarin glucoside, in the raw powders of American cockroach [50]. In their bioassay study, it was confirmed that the compound had a significant anti-inflammatory effect and significantly promoted granulation tissue proliferation. In addition, Yang et al. isolated three new dihydroisocoumarin glucosides (**12**–**14)** from *PA* together with one that was known (**15**) [51]. It was found that compound **13** could stimulate collagen production by 31.2% in human dermal fibroblasts, adult (HDFa) at a concentration of 30 μM, which indicated it should play an important role in skin repair and ulceration.

### 2.6. Alkaloids

Generally, this series of natural nitrogenous compounds can be divided into plant alkaloids and animal alkaloids, and the common alkaloids in insects are amines, imidazoles (histamine, allantoin), purines, etc.; a considerable part of them exists as toxins in the secretions (mucus) and bodies of insects. In the biogenic synthesis pathway, some alkaloids could be converted from amino acids from the family Blattidae. Most of them show anticancer, antitumor, anti-inflammatory, antiviral and antiplatelet aggregation, antiarrhythmic, and antihypertensive-cardiovascular-disease effects [52]. Therefore, they have great application potential as leading compounds and have become a hot research topic in recent years. A variety of alkaloids was identified in the n-hexane extract of the defatted products from American cockroaches—including pyrrole, piperidine, piperazine, etc.—which account for nearly 6.55% of the total mass. The researchers also obtained two quinolinone components after separation, which were identified as 3,4-dihydro-2-quinolinone and 3,4-dihydro-8-hydroxy-2-quinolinone [53]; it has been proved that the former is an inhibitor of c-AMP-specific enzymes, which can not only inhibit the aggregation of platelet and thrombosis but also enhance myocardial contractility and dilate blood vessels [54,55]. Additionally, the antennae contain octopamine, which is a type of biological monoamine related to norepinephrine structure. However, there are still many alkaloids that have not yet been discovered, and their mechanism of action must be further studied.

### 2.7. Flavonoids

As a great family of natural products, flavonoids are widely distributed in a wide range of resources. Their content in *PA* is 16.0–19.7 mg/g, which is higher than in rice locust (3.79–18.6 mg/g) and blister beetle (4.12–18.5 mg/g) [56]. This kind of component not only participates in the formation of the final color of insect bodies but also helps them resist bacteria. Due to the rapid growth and reproduction of these insects, the flavonoids in their bodies accumulate rapidly, which is different from in plant medicine and is an obvious advantage. In humans, flavonoids can scavenge free radicals in the body, have the effect of antilipid peroxidation, and promote insulin secretion, thereby lowering blood sugar levels in diabetic patients and preventing complications. Therefore, flavonoids in Blattidae are ideal lead compounds with great development potential. Two isoflavones have been found in 70% methanol extract of *PA* (see Figure 5), of which 12-(16-hydroxyl, 17, 18-dimethyl)-isoamyl-13-oxo(19,20-dimethyl)-pyro [7,8] epoxyhexenyl-Δ11, 12 3-(4’-methoxy)-phenyl-5-hydroxyisoflavone (**16**, light yellow amorphous powders, C_26_H_28_O_6_) is a new compound that has antibacterial activity against the gram-positive bacterium *Bacillus subtilis*. Generally, C_4_=O, C_5_-OH, and C_7_-OH play the major role in the related antimicrobial effects. Moreover, compound **17** is a 5,7-dihydroxy-4’-hydroxyphenyl isoflavone (light yellow solid, C_15_H_10_O_5_) which is often found in legumes and has estrogen-like functions [57]. Both of these two substances were obtained from American cockroaches for the first time. However, at this stage, there is not enough evidence to prove whether or not the new isoflavone derivative (**16**) is produced by insects from plant foods or originates from their own metabolism. Ali’s group identified and characterized the substances in *PA* by LC-MS coupled with an Agilent 6460 triple quadruple mass spectrometer and found the existence of 2,5-dihydroxy-4′-methoxy-flavanone (**18**) [58]. The compound (**18**) has a nucleus of dihydroflavonoids, the ring A can be complexed with transition metals, and C-7 of the ring A is hydroxylated to facilitate complexation, thereby enhancing its antioxidant activity. The presence of the C-44 carbonyl group can extend the conjugated system and help the flavonoids form a more stable free radical intermediate, which can result in enhanced antioxidant activity.

### 2.8. Others

Lipids are another class of macromolecular substances with important physiological functions which includes fats (triglycerides) and phospholipids, glycolipids, cholesterol, and cholesterol esters [59]. The insects in this family are full of lipids in their bodies; the feet of cockroaches can secrete lipid microdroplets with μm-level diameters, which allows them climb up and down smoothly through the capillary phenomena caused by oil surface tension. The oil on their body surface is also a type of protection suited to living in a warm and humid environment. The medicinal value of the lipid components of medicinal insects is mainly reflected by saturated fatty acids, monounsaturated fatty acids, and polyunsaturated fatty acids, e.g., oleic acid, palmitic acid, linoleic acid, stearic acid, etc. [60]. For example, GC-MS analysis showed that the oil content of *PA* mainly consisted of enol, enoic acid, alkanes, and esters. Meng et al. studied the oily chemical composition of American cockroaches and identified 19 constituents, with fatty acids accounting for 36.77% [61]. Among them, the unsaturated fatty acids were octadecenoic acid (13.86%) and octadecadienoic acid (8.23%) and the saturated fatty acid was mainly hexadecanoic acid (10.13%). Moreover, Yu et al. used SPF-CO_2_ extraction and GC-MS analysis of *PA* to obtain 50 compounds, accounting for 76.33% of the total extract [62]. The largest mass fraction is octadecenoic acid (36.48%), of which oleic acid accounts for 31.93% and (S)-11-octadecenoic acid accounts for 4.55%. In addition, there are seven compounds accounting for 29.69% of the total tested objects: (Z)-13-ethyl-octadecenoic acid ester (6.9%), 11-methyl-octadecenoic acid ester (5.86%), ethyl oleate (5.51%), ethyl linoleate (3.59%), methyl linoleate (3.49%), diisobutyl phthalate (2.23%), and ethyl palmitate (2.11%). Researchers further analyzed and compared the lipid compositions of *Periplaneta australasiae* and *PA*, and it was found that the main component types of the two insects were fatty acids and their esters, but there are some differences in the types and relative peak areas of the 21 common components [63].

Terpenoids are mainly found in sex pheromones and juvenile hormones (JH) in the family Blattidae; the former are mostly monoterpenes, and the majority of the latter belong to sesquiterpenes. As a special sex pheromone component secreted by females to attract males, blattellaquinone (**19**) was isolated from the body of *Blattella germanica* (German cockroach) and has been identified as a gentisyl quinone isovalerate [64]. JH are important hormones that control the growth and metamorphosis of insects and are secreted by the pharyngeal body. The JH of *PA* also mainly exist in female adults. The main structure of JH discovered so far is shown as compound **20** (see Figure 6), and its structural type belongs to the epoxidized farnesyl methyl ester. Various JH main chains are composed of 12 carbon atoms, but there are differences in side chains and epoxy groups [65]. The composition of sex pheromones has been extensively studied and identified as periplanone A(**21**), B(**22**), C(**23**), and D(**24**), which belong to sesquiterpenes [66].

Ecdysone is a typical representative of the steroids widely distributed in insects [67]. The major ecdysone in the American cockroach is α-ecdysone (**25**), which is shown in Figure 7. The active metabolite 20-hydroxyecdysone (Crustecdysone) of α-ecdysone plays an important role in coordinating developmental transition and sleep regulation[68]. In addition, Yin et al. isolated a sterol-like constituent (**26**) from the alcohol extract of American cockroaches [57]. The compound has cyclopentane polyhydrophenanthrene as its basic nucleus and is rich in alcoholic hydroxyl groups. Clearly, this structural character can enhance its antioxidant activity.

Additionally, dopamine, N-acetyldopamine, 5-hydroxytryptamine (5-HT) and dopamine-3-*O*-sulfate are also found in the tissues and nervous system of this family, and the secretions of abdominal glands contain p-cresol, p-ethylphenol, and p-vinylphenol [69,70].

## 3. Isolation of Bioactive Compounds

Currently, the main extraction methods include percolation, ultrasonic, and reflux with alcohol, water, or their mixture, which have the problems of losing active substances or components, low work efficiency, and complicated operation steps [71]. For insects just like cockroaches, the active components are quite complex, including both macromolecules—such as polysaccharides, proteins, and peptides—and small molecular substances—such as flavonoids, polyphenols, alkaloids, terpenes, etc. It is difficult to selectively extract target components in a single solvent extraction, and some trace valuable components cannot be obtained due to loss; even the extraction efficiency of crude extract must be further improved [72]. In addition, many of them are thermally unstable, or easily undergo oxidation and hydrolysis. Before extraction, some inappropriate pretreatment methods (such as scalding to kill them) may cause decomposition of some ingredients. Furthermore, non-medicinal parts should also be removed before extraction as much as possible. Therefore, how to extract and separate effective components from complex natural products is an important issue in accelerating the modernization process of R&D for medicinal insects. Current studies on the extraction and separation of Blattidae mainly include: (1) the effect of different solvent extracts on pharmacological activities; (2) the effect of different crude extraction methods on pharmacological activities; (3) the systematic separation process for active substances such as peptides, proteins, and flavonoids; and (4) the application of new extraction and isolation techniques/materials, which can improve current research efficiency.

### 3.1. Crude Extraction Conditions

Solvent extraction is the most common method of obtaining active substances and is based on the dissolution properties of various components and follows the rule of similar dissolving mutually. In the extraction process for the raw materials of the family Blattidae, percolation and reflux extraction are the most frequently used methods. In addition, the most popular solvent applied in the extraction method is ethanol because a large number of target ingredients in the family Blattidae have certain solubility in it.

Considering the mildness of the extraction conditions required for thermolabile constituents, a percolation method was developed in the extraction process for the peptides from American cockroaches, and ethanol with different concentrations is commonly employed as the eluent [73]. After moderate crushing, the American cockroach powders were soaked in petroleum ether for 24 h to remove fat and then soaked with 0.8 equivalents of 90% ethanol for 6 h in a percolator. After that, the defatted powders were percolated with 10 equivalents of 90% ethanol three times, and the dropping rate was 1 mL/20 s; the fresh solvent constantly extracted polypeptides through the concentration difference between the inside and outside of the cells, and the leachate flowed out from the lower part of the percolator continuously. Finally, all leachate was collected and filtered, and the filtrate was concentrated under reduced pressure at 50°C. As described above, percolation is a dynamic extraction mode with the advantages of high solvent utilization, fewer impurities, continuous operation, and protection for thermally unstable constituents.

In addition to percolation, Geng et al. employed refluxing in the extraction process of American cockroach with the index of antitumor activity and found that the optimum process was that the American cockroach powders were extracted by using 15 equivalents of 75% ethanol for 1.5 h and then soaking for 1 h. As the result, the obtained extract was found with the highest yield and the highest antitumor activity in vitro [74]. Under the above conditions, various substances in the body of *PA* can be destroyed with the further extension of the extraction time, resulting in a decrease in the yield of bioactive substances and antitumor activity. According to the bioassay in vitro, the effective parts of the crude extract were screened which were successfully obtained after the latter was extracted with a series of solvents with different polarities; the best was the part extracted using ethyl acetate. Preliminary physical and chemical reactions revealed that it contained organic acids, triterpenes or saponins, esters, lactones, coumarins, and their glycosides. Moreover, Sudha and colleagues evaluated the performance of different extraction modes (ultrasonic assistance, percolation, refluxing) through the extraction result of peptides and found that the peptide content of the extract obtained via refluxing was higher than that obtained via percolation, which could reach up to 34.18%, while the content of polypeptide obtained via ultrasonic extraction was 25.63% [75]. The results indicated that the reflux method is a better selection to obtain active peptides in the family Blattidae if the extraction duration can be reasonably controlled. *Eupolyphaga sinensis* Walker is another important medicinal insect in this family which often lives in humus-rich environment; summer and autumn are the boom seasons for its growth and reproduction. In Ge’s research [76], the antitumor effect of ethanol extract of *Eupolyphaga sinensis* Walker was tested in hepatocarcinoma-H_22_-bearing mice. The ethanol extract was also prepared using the reflux method. After extraction with 10 equivalents of 95% ethanol for 1 h, the solution was centrifuged to obtain the supernatant. Finally, vacuum drying was performed to obtain a yellow oily liquid. The fat-soluble composition obtained with this method contains a large amount of saturated and unsaturated fatty acids, which were proved to be the main substances with antitumor activity. The major constituents were palmitic acid (21.70%), cis-9-oleicacid (40.78%), cis-9, 12-linoleic acid (21.86%), cis-9-palmitoleate (9.86%), cis-9, 12, 15-linolenate (1.69%), and myristate (1.67%).

In recent years, ultrasonic extraction has also been found to be very effective in the extraction of active components from Blattidae because of its unique cavitation effect. It is generally carried out at room temperature to avoid the destruction of effective ingredients at high temperature. In the study of antitumor peptides, the powders of American cockroach were first defatted with petroleum ether. Then, the residue was soaked in 10 equivalents of 90% ethanol for 48 h and sonicated with ultrasonic power of 400 W for 20 min three times. After filtration, the filtrates were combined and concentrated under reduced pressure at 50°C. With mechanical and thermal effects, the ultrasonic radiation enhances the ability of the solvent to penetrate into the cells and accelerates the dissolution of the active ingredients in the solvent, thereby increasing the final extraction efficiency [73].

Deep eutectic solvents (DES) are a new type of solvent with properties similar to ionic liquids. As a nontoxic and degradable green solvent and an extraction solvent with universal applicability for active components, DES have the advantages of being cheap and easy to obtain, environmentally friendly, and easy to prepare, with great application value in the extraction of natural products [77]. Compared with traditional solvents, DES has a lower melting point, which is conducive to mass transfer in the extraction process, and has a wide range of polarity, which has better solubility for drug molecules and metal oxides [78].

### 3.2. Separation Conditions

Under most extraction conditions, the crude extract of the family Blattidae contains a high number of coexisting components, and the content of target substances is usually quite low. To obtain a single bioactive component with as high purity as possible, it is necessary to further separate and purify the crude extract of Blattidae. At present, many studies focus on the peptide substances with antibacterial and antitumor activities, and related purification methods for them mainly include salting-out, ultrafiltration, isoelectric point precipitation, column chromatography, etc.

#### 3.2.1. Salting-Out and Ultrafiltration

The salting-out method is often used for the separation and purification of protein with ammonium sulfate. The method of saturation of (NH_4_)_2_SO_4_ solution was used to carry out staged salting-out of the initial extract of American cockroach to obtain 0–35%, 35–70%, 70–100% crude protein [79]. The precipitate in each salting-out section was dialyzed again with Tris-HCl buffer solution (0.01 mol/L, pH 7.60), and the buffer solution was changed several times until no obvious precipitation was formed by BaCl_2_ and the target protein could be obtained. There was another salting-out method as following: a certain amount of ammonium sulfate was added to the clear concentrated solution in batches under magnetic stirring to make the final saturations of ammonium sulfate reach 25%, 50%, 75%, and 100% [80]. When the corresponding saturation was reached, the addition of ammonium sulfate was suspended and stirring was continued for 15 min to fully dissolve the salt. The resulting solution was refrigerated and allowed to stand at 4 °C for 2 h and centrifuged at 4500 r/min for 30 min to separate the supernatant and precipitate. The results of electrophoresis found that there were obvious bands in the precipitation at 25%, 50%, and 75% saturation, suggesting that sulfuric acid with appropriate saturation can precipitate the main protein from the water extract, and there were two main protein bands between 66.4–97.2 kD as a result.

Ultrafiltration membrane separation technology has been recognized internationally as one of the most promising preparative tools in the 21st century and is currently widely used in pharmaceuticals, biochemicals, water treatment, and other fields. Therefore, Wu used a hollow fiber ultrafiltration membrane with a molecular weight cutoff of 3 kD to separate small molecular peptides from the antitumor active site of *PA* and investigated the effects of rotation speed, concentration ratio, pH and operating time on the separation result [81]. Finally, the optimal process conditions were found to be speed = 4500 r/min, concentration ratio = 30%, and pH = 11 for 30 min. Moreover, Grotheer used ultrafiltration to separate the inhibitor(s) of polyphenol oxidase in German cockroach, *Blattella germanica*. Ultrafree-4 centrifugal filter units (Millipore Corporation, Bedford, MA) with the NMWL membranes of 10,000, 50,000, and 100,000 were chosen by them [82]. An amount of 1.5 mL of the sample was placed in the filter and concentrated by centrifugation at 4 °C and 7500× *g* for 35 min. Filtrates were collected and kept at 4 °C. The sample was reconstituted to 2 mL by adding 0.1 M sodium phosphate buffer (pH 6.5) and centrifuging again. As retentate on the membrane surface, the concentrated inhibitor sample was reconstituted to 2 mL by adding distilled water and assayed to show polyphenol oxidase inhibitor activity. Some examples of extraction and separation methods for the family Blattidae are shown in Table 2.

#### 3.2.2. Conventional Column Chromatography

For peptides in this family, the widely used separation materials include dextran gel, agarose gels, polyacrylamine gels, etc., with the advantages of high recovery, few interferences, mild separation conditions, and a wide range of separated molecular weights. In Bao’s research on the separation of the polypeptide extract from *PA* [73], the amount of Sephadex G-50 was 5% equivalent of bed volume and the content of peptides in the sample was 40–70 mg/mL; then, it was eluted with pure water. Finally, the collected liquid was concentrated under reduced pressure at about 50 °C and freeze-dried to obtain lyophilized powder of cockroach polypeptide.

There are also studies in which macroporous resin was used to purify the polypeptides in the crude extract of American cockroach. Compared to existing separation methods, it has the advantages of acceptable adsorption capacity, fast adsorption speed, high elution rate, and lower cost than gels. Meanwhile, it should also be emphasized that the antitumor peptides purified by macroporous resin are not always a single component, and further treatment with dextran gel or reversed-phase high-efficiency liquid chromatography preparation is needed to separate and purify the antitumor active peptides. The common types of macroporous resins include NKA-9 (polar), DM-301 (medium polar), AB-8 (weak polar), D101 (non-polar), DA201 (non-polar), and HP20 (non-polar) [83]. It was found that HP20 macroporous resin had good enrichment characteristics for the polypeptides in the extract of *PA*, and the peptide components in the extract can be effectively eluted with 70% ethanol. Additionally, HP20 resin is a nonpolar resin, mainly based on the van der Waals interaction between the hydrophobic group of the compound and the nonpolar adsorbent. During the elution with ethanol, the polarity of the eluent can be changed by adjusting the ethanol–water ratio, to enrich and purify the antineoplastic components in the extract. In addition, it was also found that the elution with a higher volume fraction of ethanol has a significant effect on the desorption rate of peptides and the inhibition of tumor cell proliferation. The best operating condition was to load 2 BV of *PA* with a mass concentration of 0.3 g/mL with a loading speed of 2 BV/h. After standing for 1 h, it was eluted with 70% ethanol (2 BV/h, 2 BV). After measuring the peptide mass fraction of the eluate, it was found that the recovery could reach 88.53%. In addition, the adsorption properties of macroporous resins of different polarities to the polypeptides from American cockroach were investigated. It was found that the separation performance of medium-polar AB-8 macroporous resin was better than that of nonpolar and polar resins, and the adsorption capacity of peptides reached 52.54 mg/g. By comparison, the adsorption capacity of other resins was lower; in particular, the adsorption capacity of HPD400 macroporous adsorption resin was the lowest at only 14.09 mg/g [86]. It should be noted not only that the large adsorption capacity of macroporous resins is required but also that the desorption efficiency should be as high as possible to ensure maximum recovery of the separated components, which is very crucial for those active ingredients with very low contents. Unfortunately, it is often the case that the amounts of some obtained components are not sufficient to complete the spectral analysis or cannot meet the needs of the subsequent activity tests.

Silica gel column chromatography is also usually used for the separation of monomer compounds, is easy to operate, and has high separation efficiency. The crude powder of *PA* was extracted with chloroform and then subjected to silica gel (200–300 mesh) column chromatography followed by gradient elution. For instance, the systems of cyclohexane-ethyl acetate (100:0~0:100, *V/V*) and ethyl acetate-methanol (100:0~50:50) were often used. After elution, six main fractions were obtained, among which two new alkaloids were obtained after further purification (3,4-dihydro-2-quinolinone and 3,4-dihydro-8-hydroxy-2-quinolinone) [53]. In a study on fat-soluble antibacterial substances of *PA*, the ethyl acetate extract (400 g) was separated on a silica gel column with dry sample loading, then eluted with petroleum ether, petroleum ether/ethyl acetate (15: 1, 8:1, 5:1, 3:1, 1:1, *V/V*), ethyl acetate, and ethanol successively (3000 mL of each eluent). After the antibacterial activity experiment, the effective components were determined to be oleic acid-1-glyceride and linoleic acid-1-glyceride [84] from the eluate of petroleum ether acetone (9:1) and 1% acetic acid. The water-soluble components of ethanol extracts from *PA* were purified using silica gel (200~300 mesh) column chromatography with the eluent of V_chloroform_:V_methanol_:V_water_ = 9:1:0.1–0:0:1 to obtain five fractions. Each fraction was subjected to repeated silica gel column chromatography, in which the third fraction was separated by column chromatography (V_chloroform_:V_methanol_:V_water_= 9:1:0.1–0:0:1) to obtain 3,6-dimethyl -2,5-piperazinedione (12 mg) and L-hydroxyproline (12 mg) [85]. Polypeptides, because of their high polarity, are usually retained strongly on the normal silica gel columns, so reversed phase silica gel (C_8_ or C_18_) columns can be used. To the contrary, the small peptides and cyclic peptides with short sequences are more suitable to be separated by normal-phase silica gel.

#### 3.2.3. Preparative Liquid Chromatography

In the past ten years, preparative liquid chromatography has been the most effective and fastest-developing purification method for natural product research. Compared to the separation achieved by using self-packed columns under low or conventional pressure, it is a more powerful method that achieves high-purity resolution through a high-load, high-resolution preparative column with characteristics of high separation efficiency and wide application range. Basseri et al. performed semipreparative reverse-phase high-performance liquid chromatography (RP-HPLC) on hemolymph with antibacterial activity to isolate and purify peptides from *PA* [28]. In total, 15 different peaks were obtained, and fractions were collected under the following conditions: flow rate = 1 mL/min; stationary phase: Spherisorb C_18_ column (250 mm×4.6 mm, 5 μm, Waters, USA); mobile phase: acetonitrile in water containing 0.1% trifluoroacetic acid (TFA), and the detector wavelength was 230 nm. As a result, two proteins with antibacterial activity were found, and the molecular weights of the proteins, measured via nonreducing SDS-PAGE, were 60 and 72 kD. A novel termicin-like peptide with the primary structure of ACDFQQCWVTCQRQYSINFISARCNGDSCVCTFRT was found in the Chinese cockroach (*Eypolyphaga sinensis* Walker) and was separated from the sample solution dissolved in Tris-HCl buffer extract with a Hypersil BDS C_18_ high performance liquid chromatography column (4.0 × 250 mm, 5 μm, Elite, China) [87]. Then, the peptide was obtained with 0.1% (*v/v*) trifluoroacetic acid/acetonitrile at a flow rate of 1 mL/min, and the absorbance of the separated fraction was detected at 215 nm (see Figure 8a).

Some Chinese scholars applied high-speed countercurrent chromatography (HSCCC) for the isolation and purification of attractants from Chinese cockroaches with a two-phase solvent system composed of *n*-hextane-ethyl acetate–methanol–water (1.5:1:1.5:1, *V/V/V/V*) [88]. The upper phase was *n*-hexane and ethyl acetate, and the lower phase was methanol and water. The flow rate, column rotary speed, and effluent detection wavelength were set at 2 mL/min, 800 rpm, and 254 nm, respectively. A Zorbax SB-C_18_ column (250 × 4.6 mm, 5 μm, Agilent, USA) was used to identify the components: 20% to 100% methanol (0–30 min); 100% methanol (30–70 min); flow rate = 0.8 mL/min; column temperature = 25 °C. The effluent was monitored at 254 nm. As the result, two new attractants were obtained from the extract samples by one-step separation and identified as (R)-3-ethyl-6,8-dihydroxy-7-methyl-3,4-dihydroisochromen-1-one and (R)-6,8-dihydroxy-3,7-dimethyl-3,4-dihydroisochromen-1-one.

### 3.3. Activity-Guided Separation

The current research on the separation and purification of *PA* is based on two methods of screening active substances. One is the tracking separation method under the guidance of activity. It is guided by the activity of the extracts using the method of screening and separating step-by-step to narrow the target range and finally identify the nature and structure of the active substance. For example, the method orientated by bioactivity is used to guide the separation of the active fractions of wound-healing of the American cockroach. First, the wound-healing activity of three different extracts of *PA* (ethanol extract, total protein, and total polysaccharides) was systematically compared, and it was found that the healing rate of the ethanol extract group was clearly higher than that of the other two groups. After that, solvent extraction and column chromatography were used to further track and concentrate the active fractions of the ethanol extract, and it was found that the potential constituents related to wound repair mainly existed in the water eluate of macroporous resin. Finally, HPLC, UPLC-MS, and semipreparative HPLC were used to further make clear the chemical composition of the water eluate. It was determined that seven compounds had wound-healing activity (two glycosides, two fatty acids, one diterpenoid, one phenolic acid, one cyclic peptide) [15]. The purpose of this separation method is to create conditions for further research on the pharmacologically active components of the family Blattidae.

The second method is the metabolism method in vivo, and the components in animal serum, urine, and bile samples before and after administration are compared. Then, further analysis of the ingredients before and after taking the medicine is conducted, and those ingredients that can be absorbed by the body are more likely to be the target active ingredients [89,90]. However, the source of the compounds identified by this method is uncertain. Meanwhile, their concentrations are very low, and a large number of disruptors coexist. They may be substances formed in the body after metabolism and absorption rather than those inherent in natural medicine. Therefore, more researchers tend to use the former as the main method of active ingredient research for Blattidae [57].

Previous experience has shown that components with high biological activity in natural products are often substances in trace amounts and are difficult to separate. Therefore, at each stage of the separation process, it is necessary to count and compare the activity and yield of each extract. Otherwise, the key active substances are likely to be lost in this process. Using the activity-oriented method to guide every step of separation, each obtained fraction can be ensured to have biological activities so that researchers have a definite direction and goal for the research and exploration of active components during separation, making it more likely that high concentrations of active fractions will be obtained [91]. Moreover, the research on the separation and purification of the components of the family Blattidae generally focuses on certain types of compounds (especially peptides), and activity guidance is very helpful in tracking the potential synergistic effects among different types. Therefore, it is necessary to conduct more in-depth research for the above problems, and some novel techniques for on-line/in situ activity assay coupled with chromatography or spectroscopy can provide useful support.

## 4. Identification and Analysis Methods

### 4.1. Spectral Features

Preliminary identification of amino acids, peptides, and proteins in the extract can be performed using biuret and ninhydrin tests. The α-amino acid in the acidic solution can react with the ninhydrin reagent, and the solution will appear purple. This method can be used for the identification and quantitative detection of amino acids from the family Blattidae. This method can also be used in ultraviolet spectrophotometry, and the maximum absorption wavelength of the obtained blue-violet substance is 570 nm [92]. Among the 20 natural amino acids, only tyrosine, tryptophan, and phenylalanine have maximum absorbances in the ultraviolet region, and their wavelengths are 275 nm, 280 nm, and 257 nm, respectively [93]. In this way, these three amino acids can be firstly identified by UV absorption spectroscopy, and the remaining amino acids can be identified by UV after color development. Biuret reagent is an analytical chemical reagent used to identify peptides and proteins and is an alkaline copper-containing solution. When the substrate contains peptide bonds, the copper in the test solution will coordinate with the peptide and the resulting complex will become purple. The concentration can be analyzed by colorimetry, and the wavelength in the ultraviolet–visible spectrum is 540 nm [94].

Due to the lack of effective chromophores, the maximum absorbance of polysaccharides from this family usually exists in the near ultraviolet region as their characteristic peak. Additionally, if there was an absorbance signal at 260–280 nm, it indicated the protein or nucleic acids could bind to polysaccharides [95]. Nucleosides have a maximum absorbance at 260 nm, while the typical UV band of proteins is often found around 280 nm, and the absorbance value at 260 nm is one tenth or lower of that of nucleosides. Therefore, when the nucleoside sample contains a small amount of protein in crude extracts of related insects, there is almost no interference with the identification results for them. The absorption peak of the hydroxyl group in the polyols in UV is below 200 nm, and the benzopyrone core in the structure of isocoumarins results in a strong absorbance peak at 311 nm with blue fluorescence. As the most easily identified type, flavonoids have two main absorption peaks, which are caused by the cross-conjugated system of flavonoids, one at 240–280 nm (band II), related to the benzoyl group of the ring A, and the other at 300–400 nm (band I), related to the cinnamoyl group of ring B (see Figure 8b) [96]. For their rapid accumulation in the insect body, these absorbances can even be observed directly by radiation of ultraviolet light on the solid samples.

In addition, infrared (IR) spectroscopy is also an important method to identify the main components in the family Blattidae. By comparing the spectra of the peptides in cockroach, *Gromphadorhina portentosa* (Blattodea, Blaberidae) to that of pure bovine serum albumin, a series of characteristic signals could be found, which indicated that peptides and proteins were the main compounds in the related samples. The strong and broad peak at 1657 cm^−1^ belongs to the stretching vibration of the C=O bond in the CO-NH group, which is the characteristic absorbance peak of the peptide bond, and the moderate intensity peak at 1535 cm^−1^ accords with the N-H in-plane bending vibration and C-N stretching vibration peaks of the amide II band. The peaks at 1541 cm^−1^ and 1403 cm^−1^ are produced by skeleton vibrations of the aromatic ring in amino acid residues [97]. In comparison, polysaccharides have many more absorbance signals in the infrared region. The characteristic peak of polysaccharides is broad and strong at 3380–3440 cm^−1^, which results from O-H stretching vibration. As a component of polysaccharides, the monosaccharides can generally be divided into furanose and pyranose configurations; generally, pyrannosides have three absorbance peaks from 1100–1010 cm^−1^, while furanosides only have two peaks in the corresponding area. From the signals in the region within 1100–700 cm^−1^ in the infrared spectrum, the configuration of the glycosidic bond and the size of the sugar ring part can be identified as follows: 890 cm^−1^ is the characteristic peak of the β-pyranosidic bond, while 840 cm^−1^ is the characteristic peak of α-pyranoside bond (see Figure 8c) [98]. The hydroxyl groups in the polyols will have a broad and strong peak at 3000–3650 cm^−1^ in the IR spectrum, and the free hydroxyl peaks appear at 3650–3610 cm^−1^, while the associated hydroxyl peaks are broader and shift to lower wave numbers. In addition, Ph-CO-*O*-C=C- is the characteristic structural unit of isocoumarins, and it can be determined from 1650 cm^−1^, 1580 cm^−1^, 1500 cm^−1^, and 1450 cm^−1^, where the benzene ring skeleton and C=O split due to conjugation. The C-*O*-C bond of lactone has two absorbance peaks in the range of 1330–1050 cm^−1^. Meanwhile, the unsaturated C-H bond on the double bond will produce a stretching vibration peak near 3080 cm^−1^. For flavonoids, their structural fragments include O-H, C-O, C=C, and benzene rings, which can be identified by the characteristic peaks at 3400 cm^−1^ (O-H), 1800–1500 cm^−1^ (benzene ring), 1714 cm^−1^ (C=O), and 1651 cm^−1^ (C=C) [99].

### 4.2. Nuclear Magnetic Resonance

Nuclear magnetic resonance spectroscopy has developed into a mature and conventional method for analyzing the three-dimensional structure of protein molecules (molecular weight ≤ 25,000). However, the NMR spectrum signals of protein molecules are very complicated. On one hand, the existence of amino acid residues leads to severe peak overlap; on the other hand, the increase in molecular weight leads to a significant increase in the transverse relaxation rate of nuclei, which makes many signals decay too quickly to detect. Therefore, the analysis and identification process of protein structure is fairly complicated, and it is often necessary to apply isotopic labeling, multidimensional nuclear magnetic resonance (including heteronuclear single quantum coherence, nuclear Overhauser effect spectroscopy), transverse-relaxation-optimized spectroscopy, etc., for identification. Due to the limited length of this article, those seeking the specific identification steps can refer to these references [100,101].

For NMR analysis of those small molecules found in the family Blattidae, it becomes simpler. (3R)-methyl-6,8-dihydroxy-7-formoxyl-3,4-dihydroisocoumarin (**5**), for example, was reported in American cockroaches for the first time in 2014; the 1H and ^13^C NMR spectra of it revealed the signals due to two phenolic hydroxyls [δ_H_ 12.59 and 12.28 (each 1H, s)], an aldehyde group [δ_H_ 10.31 (1H, s); δ_C_ 193.7], a penta-substituted benzene ring [δ_H_ 6.25 (1H, s); δ_C_ 168.7, 167.4, 149.1, 109.1, 107.6, and 100.4], an oxygenated methine [δ_H_ 4.68 (1H, m); δ_C_ 75.5], a methylene [δ_H_ 2.90 (1H, dd, *J* = 11.1, 0.7 Hz) and 2.87 (1H, dd, *J* = 3.3, 0.7 Hz); δ_C_ 35.4], and a secondary methyl [δ_H_ 1.51 (3H, d, *J* = 6.3 Hz); δ_C_ 20.9]. The relationship between all the above spectral data and structure was clear and definite, which suggested the existence of a 6,8-dihydroxy-3,4-dihydroisocoumarin core skeleton in compound **5 [49]**.

As a useful tool, ^1^HNMR is mainly applied to determine the configuration of glycosidic bonds in polysaccharides. The fully dried polysaccharide sample was dissolved in D_2_O for HNMR measurement. The proton signal of the sugar unit is mainly gathered in the range of 3.1–5.5 ppm, of which the peaks from 4.5–5.5 ppm belong to the anomeric proton region, while the peaks from 3.1–4.5 ppm belong to the C_2_–C_6_-connected proton. Due to the serious overlap of the signals at 3.1–4.5 ppm, the proton signals at 4.5–5.5 ppm are easier to distinguish and analyze. In general, the number of proton signals in this region indicates the number of glycosidic bond types. However, if the proton signals of some different glycosidic bonds overlap due to the close displacement, it is necessary to combine other evidence (such as methylation information) for analysis. The H-1chemical shift of α-formed pyranose is greater than 4.8 ppm, and the proton shift of β-formed pyranose H-1 is less than 4.8 ppm. The coupling constant (J) of the anomeric proton with its adjacent protons is also often used to determine the glycosidic bond configuration. A J of 2–4 Hz indicates an α-type sugar ring; while A J of 6–8 Hz indicates a β-type sugar ring. Compared to ^1^HNMR, ^13^CNMR has the advantages of wide chemical shift range, high resolution, and less overlap of peaks, which is often used to determine the type and configuration of sugar residues. Generally, the anomeric carbon signals of sugar rings are located at 90–110 ppm. The number of signal peaks in this range indicates how many types of glycosidic bonds there are. The chemical shift of the anomeric carbon of furanose is 103–112 ppm, and there are signals between 82–84 ppm for C-4 of aldofuranose and C-5 of ulfuranose, which are the basis for distinguishing furanose configuration from pyranose configuration [102]. For example, in the characterization of a polysaccharide from *Eupolyphaga sinensis* Walker in the family of Blattidae, the ^1^HNMR spectrum showed that the main anomeric proton signals were located at δ 4.45, 4.58, 4.76, 5.03, 5.09, 5.12, and 5.16 ppm. On the other hand, the ^13^CNMR spectrum captured signals at δ 70.07, 70.87, 71.31, 71.87, 73.00, 73.41, 74.12, 74.68, 76.31, 76.80, and 81.39 ppm, which could be attributed to the downfield shift caused by substitutions of C-2, C-3, or C-4 of glucopyranosyl residues. In addition, in the ^1^H-^1^H COSY and HSQC spectrum, correlative signals from H-1 at δ 5.03 ppm to H-2 at δ 4.07 ppm, H-2 at δ 4.07 ppm to H-3 at δ 3.93 ppm, and H-3 at δ 3.93 ppm to H-4 at δ 4.00 ppm indicated that H-1, H-2, H-3, and H-4 were located at δ 5.03, 4.07, 3.93, and 4.00 ppm, respectively. The corresponding ^13^C chemical shifts between the carbons and protons of the single bond pairs were 109.06, 82.28, 78.06, 84.9, and 62.02 ppm; hence, the glycosidic linkage was determined as α—L—Araf—(1→, which belongs to furanose [103].

In ^1^HNMR spectrum of flavonoids, the common solvent is deuterated dimethyl sulfoxide (DMSO-d_6_), and the solvent signal is at δ 2.5 ppm. For 5, 7-dihydroxyflavonoids (see Figure 9), H-6 and H-8 in the ring A appear at δ 5.7~6.9 as double peaks (*J* = 2.5 Hz) coupled in meta position. For 7-hydroxyflavonoids, there are three aromatic protons—H-5, H-6, and H-8—in the ring A. H-5 is coupled to H-6 adjacently with a double peak (δ 8.0, d, *J* = 8.0 Hz), and the coupling between H-8 and H-6 also shows a double peak (δ 6.3~7.0, d, *J* = 2 Hz). H-6 is a double peak (dd, δ 8.0, *J* = 2 Hz) because it is coupled to the adjacent position of H-5 and at the same time coupled to the interposition of H-8. In 4’-oxygen-substituted flavonoids, the four protons in the ring B can be divided into two groups—H-2’ and H-6’ and H-3’ and H-5’—and each group of hydrogen is a double peak (2H, d, *J* = 8.0 Hz) with the chemical shift of δ 6.5–8.1, which is in a slightly lower field than that of the protons in ring A. H-3 in the ring C of flavonoids shows a sharp single peak at δ 6.3–6.8. The signal of H-3 on the C ring of flavonoids often appears as a sharp single peak at δ 6.3–6.8, while H-2 in the isoflavone is affected by oxygen atoms and carbonyl groups and shows a sharp single peak in the lower field. For example, the ^1^HNMR spectrum of the isoflavone named as 12-(16-hydroxyl, 17, 18-dimethyl)-isoamyl-13-oxo(19,20-dimethyl)-pyro [7,8] epoxyhexenyl-Δ11, 12 3-(4’-methoxy)-phenyl-5-hydroxyisoflavone (**16**) showed two adjacent dd-splitting peaks at δ_H_6. 66 (dd, 2H, *J* = 8.4 Hz) and 7. 04 (dd, 2H, *J* = 8.4 Hz); δ_H_7. 22 (dd, 1H, *J* = 6.6 Hz) and 7. 28 (dd, 1H, *J* = 6.6 Hz). In addition, there is an isolated hydrogen signal at δ_H_ 7.27. The above hydrogen spectrum characteristics reveal that the compound is a flavonoid compound [57].

Finally, the chemical shift of the active hydrogen on the hydroxyl group in the typical structure of polyols is affected by the conditions (concentration, temperature, solvent) in ^1^HNMR, and the chemical shift will change in a range of 0.5–5.5 ppm with a blunt peak. The heavy water exchange method can identify the absorption peak of active hydrogen because the peak of active hydrogen disappears after adding deuterated water [104].

### 4.3. Chromatography–Mass Spectrometry

Liquid chromatography–mass spectrometry (LC-MS) is known for its high efficiency, good resolution, wide peak capacity, and ultra-high sensitivity. There was a successful example in which Ali et al. used LC-MS to identify compounds present in cockroach lysates [58]. Firstly, the cockroach brain, hemolymph, and muscle lysates were extracted with water and methanol, respectively, and then the related samples were qualitatively analyzed using LC-MS. A Merck C_18_ column with a particle size of 3 μm (5.5 cm × 4.6 mm) was used for liquid chromatography separation at 25°C; 90% solvent A (0.1% formic acid in Milli-Q water) and 10% solvent B (0.1% formic acid in MeOH) was used as the mobile phase, and the flow rate was set to 0.6 mL/min with a linear gradient as follows: 10% solvent B for 4 min, 80% solvent B for 3.2 min, and 10% solvent B for 2.8 min. Lastly, electrospray ionization (ESI) + jet stream ion mode on the triple quadrupole analyzer was applied in the identification of target compounds. The data obtained from the LC-MS for each part (extract of brain, hemolymph, or muscle lysates) contained over 160 peaks for water-soluble and over 170 peaks for methanol-extracted lysates, which contained 2,5-dihydroxy-4′-methoxy-flavanone (**18**), 2-ethyl-2-methoxypentanoic acid, 7-(isopropoxy)-2,2,5-trimethylchromene, 5-methyl-2-(thiophen-2-yl)pyridine, 2-hydroxy-2-methylhexanoic acid, and 2,7,12,17-tetraethyl-3,8,13,18-tetramethyl-21*H*,23*H*-porphine copper. Some of these compounds identified via LC-MS have been confirmed to have biological activity (i.e., the antibacterial and antioxidant effects of flavanones and the antitumor effects of sulfonamides and furanones), but the majority of compounds in the research remained unidentified or possess main biological activities that are not yet known. Therefore, a detailed characterization of all compounds is further needed.

### 4.4. Gel Electrophoresis

Gel electrophoresis is mainly used for the identification of peptides, including SDS-PAGE (sodium dodecyl sulfate polyacrylamide gel electrophoresis) and western blotting. During electrophoresis, the mobility of a peptide depends on its net charge and the size and shape of the molecule. After SDS is added to eliminate charge factors as an anionic detergent, the electrophoretic mobility depends only on the size and shape of the molecule, so the molecular weight of the protein can be determined. To analyze the molecular weight of proteins in *PA* via SDS-PAGE [73], the relationship between the concentration of the separation gel and the molecular weight of the protein separation was explored. The separating gel (12%) in the polyacrylamide gel plate was composed of 30% acryl, 1.5 M Tris-HCl, 10% SDS, 10% ammonium persulfate, TEMED, and H_2_O. Stable flow electrophoresis was used for sample addition with 10 μL of each test sample and 5 μL of the marker. After the bromophenol blue indicator entered the separation gel during electrophoresis, the current was adjusted to 30 mA. The experiment found that the mixed protein in the ethanol extract of *PA* diffused during electrophoresis, which may be caused by denaturation of the macromolecular protein exposed to high concentration of ethanol and high temperature during the preparation of the extract. In addition, the proteins were rich in the sample solution of flash water extract of *PA*, and the molecular weight of the most concentrated protein was 70–100 kD and about 5 kD with clear bands, indicating that the macromolecular protein components are suitable for the low-temperature extraction of aqueous solutions. Almost no macromolecule was observed in the ethanol-extracted samples using reflux of *PA*, indicating that it only contained a very small amount of protein and that the rest was composed of peptide mixture with a molecular weight of less than 15 kD, which proved that high concentrations of ethanol make it difficult to extract large molecules like proteins.

Western blotting can also be used to identify proteins and peptides. It is a hybrid technique that combines high-resolution gel electrophoresis and immunochemical analysis techniques. It has the advantages of large analysis capacity, high sensitivity, and strong specificity and is one of the most commonly used methods for the detection of protein characteristics and distribution, quantitation of tissue antigens, quality determination of polypeptide molecules, and so on. The principle is to use polyacrylamide gel technology to efficiently separate biologically active substances. The separated sample can be transferred to another solid phase carrier, usually using cellulose acetate film. The solid phase carrier adsorbs proteins in the form of non-covalent bonds and can maintain the type of polypeptide separated by electrophoresis with its biological activity unchanged. The protein or peptide on the solid phase carrier is used as an antigen to react with the corresponding antibody. Then, it reacts with an enzyme or an isotope-labeled secondary antibody and undergoes substrate color development to detect the protein component expressed by the specific target gene separated by electrophoresis [105]. Studies have used this method to investigate the allergen content of German cockroach (*Blattella germanica*) whole body (GWBE) and fecal (GFE) extracts [75,106]. In western blotting, approximately 20 μg of protein per lane of GWBE and GFE was electrophoresed in 15% polyacrylamide gel. Proteins in the gels were then electrophoretically transferred to CNBr-activated nitrocellulose and stained with India ink (1:1000 dilution with PBS–0.05% polysorbate 20). Western blot analysis revealed 20 protein bands in GWBE and 22 protein bands in GFE that bound IgE from skin-reactive positive sera. In GWBE and GFE, 81% and 76% of the sera bound to the 60 kD band, and 60% and 54% to the 67 kD band, respectively. In summary, this investigation identified that German cockroach feces possess significant allergenic activity, and the allergen bands at 60 and 67 kD showed the most significant reactivity.

## 5. Pharmacological Effects and Structure–Activity Relationship

### 5.1. Antitumor Effect

Increasing evidence has revealed the antitumor effects of the family Blattidae on a variety of cancer cells [107]. It can be used in all stages of tumorigenesis by inhibiting the synthesis of DNA, RNA, and proteins and preventing the energy metabolism of tumor cells. Specifically, the extract of the family Blattidae mainly exerts antitumor activity through five aspects: cell cycle arrest, induction of apoptosis, repression of tumor gene expression, antiangiogenic effect, and reversal of drug resistance. At present, the chemical components with antitumor activity of the family Blattidae which have been reported include sulfated glycosaminoglycans (GAGs), chitosan, peptides, polysaccharides, coumarins, and chalcone. As a highly sulfated linear polysaccharide in the glycosaminoglycans family consisting of repeating disaccharide units, Heparan sulfate (HS) is composed of *N*-acetylglucosamine (GlcNAc) and glucuronic acid (GlcA) residues (see Figure 10). It regulates tumor cell proliferation by binding to fibroblast growth factor 2 (FGF-2). The negatively charged sulfuric acid group in HS and the positively charged arginine and lysine in FGF-FGFR (fibroblast growth factor receptor, FGFR) are combined through electrostatic interaction and hydrogen bonding, so that HS and FGF-FGFR form a ternary complex. This combination allows HS to regulate cell growth by controlling the release of FGFs, thereby forming a negative regulatory mechanism. Therefore, the binding of FGFs to HS requires that their ends have sulfation sites, such as N-SO_3_, 6-*O*-SO_3_, and 2-*O*-SO_3_, and the degree of sulfation on HS is the key to its regulatory effect [108,109]. As an alkaline polysaccharide, chitosan has a similar structure to HS and a good antitumor effect.

Among the coumarins, 4-methyl-7,8-dihydroxycoumarin (DHMC) shown in Figure 10 can cause non-small-cell lung carcinoma cells (NSCLC) apoptosis without any toxicity to normal cells, which is superior to the currently marketed drugs (doxorubicin, 5-fluorouracil, cisplatin, etc.) for the treatment of non-small-cell lung cancer [110,111]. In the structure–activity relationship study of coumarins acting on U-937 cells, the aromatic portion of the coumarin nucleus is of major importance for differentiation-inducing capacity. Specifically, ortho-dihydroxycoumarins (o-DHC) proved to be more effective apoptosis inducers than monohydroxycoumarins or open-chain analogues, confirming that the presence of the catechol moiety is essential to induce apoptosis in U-937 cells. In addition, although the δ-lactone portion does not exert biological activity by itself, it could increase the potency of the growth inhibitory activity of the catechol moiety on U-937 cells [113]. The existence of these structural fragments makes o-DHC more toxic to tumor cells than normal blood cells. The antitumor mechanism of coumarins is shown in Figure 11. Furthermore, ortho- or meta-dihydroxycoumarins have more cytotoxic effect on human tumor cell lines than mono-hydroxycoumarins. It should be noted that the integrity of the coumarin nucleus plays an important role in the biological activity of o-DHC to promote apoptosis [114]. The introduction of trifluoromethyl groups in the coumarin structure will increase fat solubility and easily enter tumor cells. Therefore, the antitumor activity of fluorocoumarin compounds is stronger than that of simple coumarin compounds.

### 5.2. Antifibrosis Effect

Liver fibrosis is the response of damage repair to excessive deposition of liver extracellular matrix caused by various chronic pathogenic factors. Recently, many pharmacological experiments have explored the mechanism of its anti-liver-fibrosis process, and the results indicate that it may be related to the anti-lipid-peroxidation reaction and the reduction of liver fibrosis cytokine expression. Clinical practice has confirmed that “Ganlong” capsule consisting of the extract of *PA* has a significant inhibitory effect on chronic hepatitis B. The main ingredient of glycosaminoglycans has a significant preventive effect on chronic alcoholic liver injury in rats. The mechanism of action may be related to the anti-lipid-peroxidation reaction and the reduction in the level of inflammatory factors in the body. Glycosaminoglycans can increase the activity of superoxide dismutase (SOD) and glutathione (GSH), thereby reducing the production of lipid peroxidation products and avoiding the increase of liver cell membrane permeability and the damage of mitochondria in liver cells. In addition, this ingredient has been proven to significantly reduce the concentration of TNF-α in rat serum, which leads to liver protection by delaying the inflammatory response caused by alcohol in liver cells [115].

Polysaccharides are the main medicinal substances in the family Blattidae, with antifibrosis and liver-injury-preventative effects. Generally, D-glucan connected by β-(1→6) has lower activity, while polysaccharides with β-(1→3)-D-glucan show strong biological activity, but there must be a certain amount of β-(1→6) bonds on the branch [116]. There are four types of high-level structures of active polysaccharides: type A is a stretchable ribbon, type B is a buckling spiral, type C is a wrinkle ribbon, and type D is a curved line graph. Polysaccharides with type B structure can enhance immune function, type A is less active, and types C and D are generally inactive [117]. The content of uronic acid in the polysaccharide is closely related to the ability of free radical scavenging and antioxidant activity. The higher the content of uronic acid in polysaccharides, the stronger the ability to activate the hydrogen atoms on the anomeric carbon to undergo redox reactions with oxygen free radicals [118]. The antioxidant capacity of polysaccharides is also affected by the composition of monosaccharides. Polysaccharides rich in Ara, Rha, and Gal as monosaccharides show strong antioxidant activity. Meanwhile, the content and the position of sulfate in polysaccharides affect their antioxidant activity. Higher sulfate content in polysaccharides leads to higher activity because the presence of sulfate activates the hydrogen on the anomeric carbon, which enhances the hydrogen supply capacity of the polysaccharide. In addition, sulfation at C-4 and C-6 positions of the polysaccharide has strong antioxidant activity, while the sulfation at C-2 inhibits this activity [119].

### 5.3. Wound-Healing Effect

Common injuries include acute injuries, burns, and ulcers. Wound healing is a series of complex biological processes involving multiple cells, extracellular matrix, and cytokines, which are mainly composed of bleeding, inflammatory, proliferative, and remodeling types. The mechanism of action to promote wound healing includes (1) network interaction regulation between various cells and cell growth factors on the wound; (2) strengthening the oxidative metabolism function of immune active cells on the wound; (3) improving the microcirculation of the wound; (4) keeping the wound moist, and so on. Any abnormality at each stage can lead to delayed wound healing. Although some growth-factor-based therapies have shown a powerful effect in accelerating wound closure, due to the complex compatibility of multiple growth factors involved in wound healing and the possibility of carcinogenesis, their safety has always been the focus of attention. Thus, there is a real need for an alternative to synthetic wound-healing products, and natural products are the most reliable and successful sources of drug leads in this aspect.

In current research, Zhu [15] studied the wound-healing activity of different solvent-eluted components of *PA* and found that the healing rates of ethanol extract (EE) are the highest compared to the total polysaccharides (TPS), total proteins (TP), and negative control (NC) groups. Specifically, the healing rate of the EE-treated group significantly increased to 65% at the third day compared to only 16% in the NC group. Seven compounds (including cyclopeptides, diterpenoid, phenolic acids, fatty acids, and glycosides) were identified from this fraction with the UPLC-MS method, among which the diterpenoid, one phenolic acid, and two glycosides were first reported in *PA*. It was found that the high content of arbutin in the active part will lead to stimulated collagen production that is closely related to skin and ulcer repair. In addition, arbutin can significantly reduce proinflammatory cytokines, including IL-1β and TNF-α. The active part also contains diterpenoids and phenolic acid compounds, which are rich in phenolic hydroxyl groups. Therefore, it has obvious antioxidant activity, which can eliminate free radicals related to inflammation and promote healing.

The isocoumarin glycoside (**13**) in the extract of *PA* can stimulate collagen production by 31.2% in human dermal fibroblasts adult (HDFa) at a concentration of 30 μM, which indicates that it plays an important role in skin repair and ulceration. Isocoumarin nucleus is the most key active structure. It was found that when the basic structural nucleus was destroyed, the activity of promoting collagen production would be significantly weakened [49]. Moreover, the active ingredient in ethanolic extract contains epidermal growth factor (EGF), which can promote the proliferation of granulation tissue and mediate the mucosal repair effect of epithelial cells. The polyol component in *PA* can significantly promote the growth of granulation tissue [15], but its concrete structure must be determined by further research.

### 5.4. Anti-Inflammatory Effect

As another meaningful activity, the anti-acute-inflammation effect and mechanism of American cockroach extract CII-3 were investigated using in vivo models; the results showed that CII-3 could significantly reduce the swelling of the auricle model induced by dimethyl benzene and then lower the content of PGE2, histamine, and malondialdehyde (MDA) in the inflammatory part while increasing SOD activity [120]. Histamine stimulates smooth muscle to contract, the relaxation of capillaries, and increase in osmotic pressure of blood vessel walls through the H1 receptor effect, which is one of the mechanisms of inflammation. Sesquiterpenes are the main active substances with anti-inflammatory effects. The lactone ring and extra-ring double bonds in sesquiterpene are essential groups for its anti-inflammatory activity (see compound **27**). In particular, the α-methylene-γ-butyrolactone moiety was essential for the anti-inflammatory effect of TNF-α secretion in activated macrophages. Once the lactone ring or the double bond outside the ring is changed, the activity will be significantly reduced. After 1-hydroxyl group in compound **28** (Figure 10) disappears, the anti-inflammatory activity can be significantly enhanced. At the same time, it was found that the activity of the 1-hydroxyl group had no obvious change after acetylation [112].

### 5.5. Antibacterial Effect

Insects of the family Blattidae can carry more than 50 type of bacteria without infection, and the ability to do so is closely related to the antibacterial substances in their bodies. For instance, the antibacterial peptides isolated and purified form the body of *PA* are resistant to gram-negative and positive bacteria, which also have a broad-spectrum antibacterial effect. Antibacterial peptides first act on the outer wall of bacteria, causing depression and perforation in the outer wall. Eventually, the substances in the bacteria were leaked, and then the bacteria disintegrated [28]. Ali’s group found that the crude brain homogenate extract (100 μg/mL) of *PA* had a potent bactericidal effect on methicillin-resistant Staphylococcus aureus (MRSA) and pathogenic Escherichia coli K1, achieving a bactericidal effect of more than 90% [58]. Body fat and muscle lysate exhibited no bactericidal activity against MRSA and E. coli K1 at the same concentration, while hemolymph showed bactericidal effects of 35% and 20% against the above two bacteria, respectively. Among them, the relative molecular weight of the antimicrobial peptide was less than 10 kD, and it did not show toxicity to human cells.

From the perspective of the structure–activity relationship, the antimicrobial peptide is first attached to the surface of the bacterial membrane due to electrostatic attraction. Then, the hydrophobic C-terminus is inserted into the hydrophobic region of the membrane and changes the conformation of the membrane, and a number of antimicrobial peptides form ion channels on the membrane, leading to the escape of certain ions and the death of bacteria [121].

### 5.6. Others

In addition, the family Blattidae also has myocardial protection and antioxidant effects. The clinical research showed that the “Xinmailong” injection, with the main active ingredients of adenosine, inosine, protocatechuic acid, and pyroglutamate dipeptides from *PA*, had various effects on the cardiovascular system, such as increasing myocardial contractility, reducing pulmonary artery pressure, and dilating blood vessels, with good effect on congestive heart failure. It can promote Ca^2+^ inflow of myocardial cells and lastingly increase the positive muscle strength of the heart. At the same time, related constituents can also expand the coronary arteries, increase blood flow, and inhibit myocardial damage mediated by oxygen free radicals. In addition, it has the effect of expanding the blood vessels of the lungs and kidneys and reducing the pressure of the body arteries [122].

Cancer, aging, and other diseases are mostly associated with excessive free radical production. Different extracts and components of the family Blattidae can express scavenging activity against different oxidation systems. Polysaccharides such as sticky sugar amino acid and heparan sulfate in *PA* have strong scavenging abilities of hydroxyl free radicals (•OH) and have a certain protective effect against cell damage from hydrogen peroxide. High-dose *PA* oil (800, 1000 μg/mL) showed a significant protective effect on the oxidative damage of SH-SY5Y cells caused by H_2_O_2_ due to the increased activity of cell-antioxidant enzymes SOD and GSH-Px and the decrease in lipid peroxide MDA content. The cell viability increased along with the oil concentration of *PA*, with the highest cell viability of 69.49% at the concentration of 1000 μg/mL. The viability of cells not treated with the oil was 51.69%, while that of the positive control group (VE) was 76.51%. Therefore, it possesses the potential to treat diseases related to oxidative stress in vivo, such as cardiovascular and nervous system diseases [123]. Some examples of the pharmacological activities of the family Blattidae are shown in Table 3.

## 6. Key Issues in Basic Study and Drug Development

Among the pharmacological activities of the family Blattidae, antitumor and tissue repair are attracting more attention from both academia and industry. The insects in this family have strong fecundity and can reproduce in four seasons, and their ideal living temperature is in the range of 24–32 °C. Once the female insects mate, they can keep producing eggs for their whole lives. A fertilized female cockroach can produce hundreds of thousands of larvae in one year with enough food and can produce more than three generations asexually. This provides great convenience for their captive breeding. Once the yield and quality are stable as well as uniform, they can be used as pharmaceutical raw materials on a large scale. At the same time, due to the lack of comprehensive and systematic research on the aforementioned lead compounds and active fractions, current patent medicines and health products are mainly based on the crude extracts or refined powders of related insect bodies.

As a result, the preparations of family Blattidae in China are commonly applied in clinic including “Kangfuxin” liquid, “Xinmailong” injection, “Ganlong” capsule, and “XiaozhengYigan” tablet, and related production enterprises have their own breeding base. Particularly, Kangfuxin liquid has been approved by the State Food and Drug Administration in China for clinical wound treatment. The preparation is obtained from the ethanol extract of *PA* with the main components of polyols and peptides. Over the years, Kangfuxin liquid has been widely used in the treatment of wounds, pressure ulcers, peptic ulcers, oral ulcers, and chronic gingivitis with the characteristics of excellent effectiveness, convenient administration, and no serious adverse reactions [5,129,130]. Xinmailong injection has been applied in the therapy of heart failure (HF) in China, which contains four active ingredients: adenosine, inosine, protocatechuic acid and pyroglutamine dipeptide. The extensive usage for HF patients has proved the effectiveness of Xinmailong injection in dilation of coronary arteries, enhancement of myocardial contraction and inhibition of ventricular remodeling, with few adverse reactions [131,132]. In addition, Ganlong capsule is a natural prescription drug extracted from *PA*, which is mainly composed of sticky sugar amino acid. Since it came into the market in 2006, the widespread clinical application has proved that it has a satisfied therapeutic effect on chronic hepatitis B and liver fibrosis, as well as immune regulation and liver protection with little side-effect [133]. Therefore, it is very beneficial to exploit the family Blattidae, which is a rich medicinal resource with great potential for medical applications. In this process, a series of key problems must be considered, such as the screening and structure optimization of lead compounds, medicinal mechanisms, preparation process, and quality control.

### 6.1. Pharmacodynamic Components Research

As far as the research of medicinal substances of the family Blattidae is concerned, at this stage, fewer specific pharmacologically effective ingredients have been found, and fewer studies have been able to clearly indicate what kind of substance is the effective ingredient. Secondly, in addition to the main active ingredients, such as proteins, peptides, and amino acids, the analysis of other chemical constituents in the family Blattidae and their pharmacological activities are relatively weak. Many of the components found so far are not completely clear about their biological activities together with the joint actions among them. In addition, there are reports that after the family Blattidae is used as crude powder with or without removing oil, it has certain toxic side effects on patients with liver dysfunction [134], but the specific toxic substances causing this antagonistic effect must be further explored. Therefore, it is essential to carry out in-depth basic research for the family Blattidae to find effective components of drugs with therapeutic actions and clarify their toxic sources. Moreover, if some parts (e.g., head, feet) of the insect do not have active ingredients but may trigger drug safety issues, then these parts should be removed in the pharmaceutical process. At the same time, the influence of species, source, sex, age, and growth environment of insects on the pharmacodynamic effects is also worth studying. Lastly, the potential impact of the operating conditions of each production section on the effective substances should also be minimized, and the current unreasonable process parameters need to be improved according to the actual pharmacodynamic effects. These are all key issues with practical significance.

### 6.2. Clinical Assessment and Mechanism Research

The research on the mechanism is not only to explain how the drug exerts its curative effect but also—more importantly—to discover the key targets, pathways, and links in the process of drug onset. It could help to develop more new drug varieties and find suitable clinical medication methods. The mechanism of activities has not been clearly investigated. For example, the antibacterial activity of the family Blattidae mainly focuses on the research of drug efficacy, and there is less research focused deeply on its mechanism. Meanwhile, a series of pharmacological effects, such as antihepatic fibrosis, antioxidation, anti-inflammatory, tissue repair, and other mechanisms of action, have not yet formed a systematic network, and the key mechanism of the family Blattidae must be further clarified.

In a previous study, the Lewis lung cancer model was applied in mouse and human peptic carcinoma cell line BGC-823 to test the antitumor effect of “Kangfuxin” liquid (the extract of *PA*) [124]. The results revealed that it could inhibit the proliferation of tumor cells and significantly reduce the number of S-phase cells. For the Lewis lung cancer model in mice, the cells were blocked at the G_0_/G_1_ stage, and the cells in the human peptic carcinoma cell line BGC-823 were blocked at the G_2_/M stage. Moreover, it was also found that there are fewer microvessels around the tumor tissue under the action of the extract of *PA* and that the possible mechanism was related to the inhibition of angiogenesis [125]. When the apoptosis of human hepatoma cells SMMC-7721 was induced by the extract of *PA*, it was found that the membrane potential of mitochondrial was continuously decreasing with the loss of activity. Meanwhile, the activities of Caspase-3 and Caspace-9 gradually increased, which was an early indication of cell apoptosis [126]. When the protective effects of different extracts of *PA* (water extract, alcohol extract, and insect powder suspension) on acute immunological liver injury caused by concanavalin A (ConA) in mice was studied, and it was found that the levels of malondialdehyde (MDA) in the liver tissues of mice decreased in varying degrees under the action of these extracts. Meanwhile, the levels of superoxide dismutase (SOD) and glutathione (GSH) in tissues increased significantly, which could effectively reduce liver pathological damage in mice. The results indicated that the antifibrosis activity resulted from the antilipid peroxidation reaction and the scavenging of free radicals in the liver. Among these, the treatment effect of the water extract and insect powder suspension was significantly better [127]. As a well-known preparation of *PA*, “Kangfuxin” liquid has been used in the treatment of various traumas, ulcers, and bedsores. A large amount of clinical data shows that it has a good effect on gastric ulcers, peptic ulcers, ulcerative colitis, burns and scalds, and war wounds, with few side effects. Recently, many scholars have devoted themselves to explaining its pharmacological mechanism, and the results indicate that it may be involved with multiple actions at different stages. The main effect includes four aspects: regulating the immune function of the wound, promoting the proliferation of granulation tissue, increasing the formation of wound blood vessels, and promoting the expression of wound growth factors.

Moreover, “Kangfuxin” liquid can significantly increase the number of wound inflammatory cells and activate the immune regulation function of the wound. The number of neutrophils increased at the first stage, and then the number of macrophages and lymphocytes increased. The preparation can also improve the spontaneous and chemotactic functions of neutrophils so that they can quickly enter the wound for phagocytosis, remove necrotic tissue, and initiate repair [90]. Tang et al. proved that ML-A2 (the water extract of *PA*) had the strongest scavenging effect on 1,1-diphenyl-2-picrylphenylhydrazine (DPPH) free radicals with an IC_50_ of 0.311 g/L. In addition, the extract ML-A4 showed the strongest inhibitory effect on spontaneous lipid peroxidation of rat liver tissue homogenate, and the inhibitory rate was found to be above 70% [123].

Traditional Chinese medicine has the characteristics of being multicomponent and multitarget and of having coordinated integration with the synergistic action of multiple ingredients [135]. The preparations of the family Blattidae are the final form in clinical application, and it is the core of the direct effect on disease and clinical efficacy. According to the different treatment purposes, the relationship between the ratio of ingredients and components within the preparation changes accordingly, and a certain reasonable composition is formed in the final applied preparation, thereby producing different therapeutic effects. In the study of mechanism and pharmacodynamic components, the optimization of the quantity–ratio relationship is carried out based on the research of the mechanism of action, and then the key effective ingredients or ingredient groups with clear composition, specific mechanisms, and reliable drug efficacy can be found. Such research methods help to further understand the relationship between the constituents and efficacy of the family Blattidae from the perspective of medicinal chemistry.

### 6.3. Quality Control Research

At this stage, the qualitative detection and quantitative analysis of certain index components are still the main content of the quality evaluation method of the family Blattidae, their preparations, and their intermediate extracts, such as the determination of total amino acid content via spectrophotometry [136]; the determination of amino acid content via pre-column derivatization [137]; or the optional determination of uracil, hypoxanthine, and inosine. The chemical composition of the family Blattidae is complex, and its medicinal ingredients do not work alone but play a collaborative role as a whole according to the natural and reasonable proportioning relationship. Therefore, it is not reliable to rely on single (or a few) component indicators to represent various active fractions from the crude extract, and it is necessary to further improve quality control methods and improve existing quality assurance standards.

Specifically, it is necessary to introduce more reasonable standard constituents for the profile effect study of the family Blattidae. For example, high performance liquid chromatography and other methods are supposed to analyze the characteristic standard substances in the family Blattidae (such as peptides, proteins, polysaccharides, nucleosides, etc.) to improve quality standards. At the same time, to ensure the quality of medicinal materials, the cultivation should be carried out in the breeding process according to the Good Agricultural Practices (GAP) standard, and the production area and harvest period should be fixed to guarantee the standardization of cultivation. In addition, it is necessary to formulate identification standards and establish comprehensive fingerprints of the family Blattidae. As is known to all, the fingerprint is a unique “identity card” for medicinal materials [138]. Chromatography is the most widely used method, of which high-performance liquid chromatography is the most widely used. A Diamonsil-C_18_ chromatographic column (250 mm ∗ 4.6 mm, 5 μm) was used on Agilent high performance liquid chromatograph to detect 10 batches of American cockroaches from different origins. The mobile phase was acetonitrile-0.1% trifluoroacetic acid aqueous solution, the detection wavelength was 256 nm, the column temperature was 25 °C, the flow rate was 0.8 mL/min, the running time was 100 min, and the injection volume was 7 μL. The results showed that the fingerprints of *PA* from different origins had 22 common peaks, with an average similarity of 0.982, suggesting the high similarity of the chemical components. There were 10 large and stable common strong peaks in the fingerprint spectrum, and the sum of their peak areas accounted for more than 85% of the total peak area. The representative peaks are peak 3 uracil (4.95%), peak 4 hypoxanthine (10.65%), and peak 11 inosine (17.03%). Additionally, electrophoresis is popularly used to analyze medicinal materials rich in protein and peptides and is also highly specific and sensitive, so it is suitable for the quality control of American cockroach and similar insects.

It should be noted that based on the characteristics of the family Blattidae that are similar to biological products, modern biotechnology and research methods should be introduced from the perspective of biological activity when evaluating internal quality. For example, the quality evaluation model based on biological effects is used to evaluate the quality and toxicity of the family Blattidae. The application of new technologies—such as proteomics, transcriptomics, metabolomics, biochip technology, TLC-bioautography, high-content analysis, and other biological evaluation studies—should be based on new emerging research methods and means; a new quality control method led by multi-index evaluation and supplemented by chemical analysis, fingerprints, and quantitative determination should also be established to make up for and improve the existing quality control system so that the safety and effectiveness of medicines can be guaranteed and the modernization and industrialization of the family Blattidae can be promoted.

## 7. Conclusions

Recently, the new discoveries and uses of valuable substances from the family Blattidae have been continuously reported (such as anticancer, skincare, hair-generating, biomaterial, medical addressing, etc.); in many cases, however, the corresponding material basis is not clear, the preparation method is not advanced enough, and the systematic utilization together with comprehensive development of the entire resource is far from reaching the ideal level. For the first time, this review comprehensively summarizes the modern medicinal research of the family Blattidae from the perspectives of chemical components and separation and purification methods together with their identification. The key issues in the study of the family Blattidae, including chemical profile, mechanism, and quality control, were summarized and analyzed to provide a reference for its further development and clinical application. There have been many successful precedents for the development of the medicinal value of this family, and the pharmacological activities and clinical efficacy have been basically confirmed.

At present, there is some research on many aspects, such as the phytochemical investigation of polysaccharides, nucleosides, polyols, peptides and proteins; isolation and identification methods; and their physiological functions and mechanisms. However, whether or not it has other biological activities that have not been discovered by scholars must still be further studied. With the advancement of science and technology and the continuous improvement of research methods, the discovered basic pharmacological effects will become increasingly clear. Finding a method of improving the targeted precise separation of lead compounds of the family Blattidae has become a challenge. To solve this problem, a series of highly selective enrichment and purification techniques can be applied under the guidance of spectral analysis and bioactivity. In addition, new green solvents (such as ionic liquids, eutectic solvents) can also be used for the extraction of active ingredients from the family Blattidae, which is a biodegradable and biocompatible material. Moreover, the exploration of chemical modification and structure–activity relationships of the discovered constituents must be strengthened, and it is also a trend to further clarify the difference of species and medicinal parts (head, body, wings, shell, antennae, and feet—even secretions) of the family Blattidae and to elucidate the mechanism of action of various biological activities and their coordinated effects at the molecular level. It is believed that in the near future, people will have a completely new understanding of this traditional medicinal pest and will continue to discover its potential pharmaceutical value in development.

## Figures and Tables

**Figure 1 molecules-27-08882-f001:**
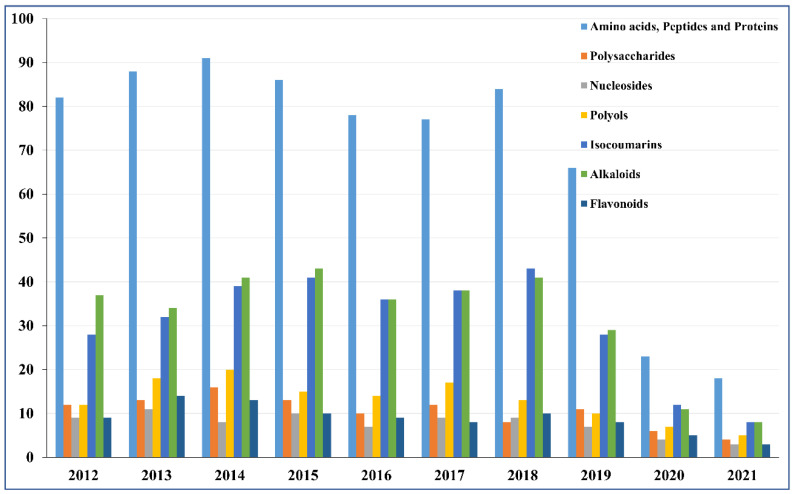
The summary of the number of works on different chemical components in the family Blattidae in past years.

**Figure 2 molecules-27-08882-f002:**
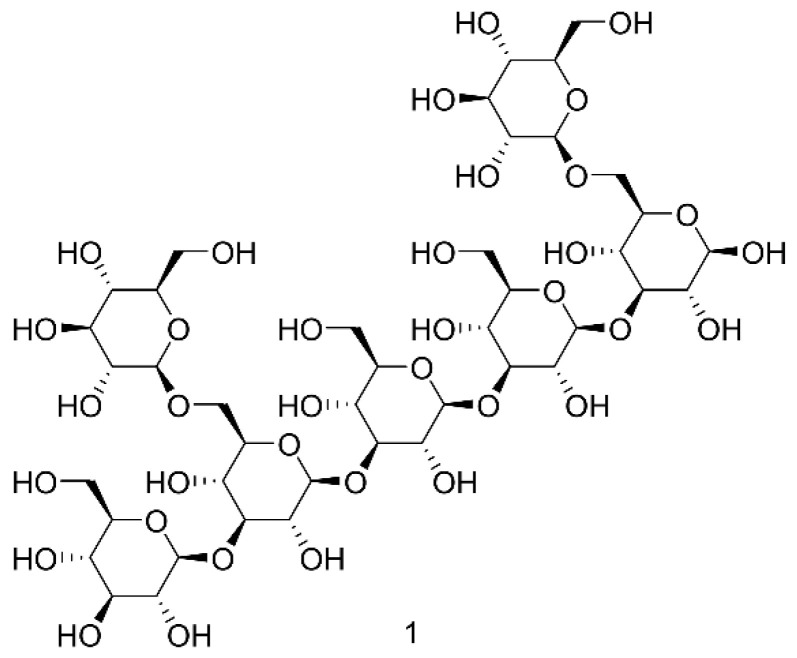
β-(1-3)-D-glucans with β-(1-6) single glucose residue side chains [35].

**Figure 3 molecules-27-08882-f003:**
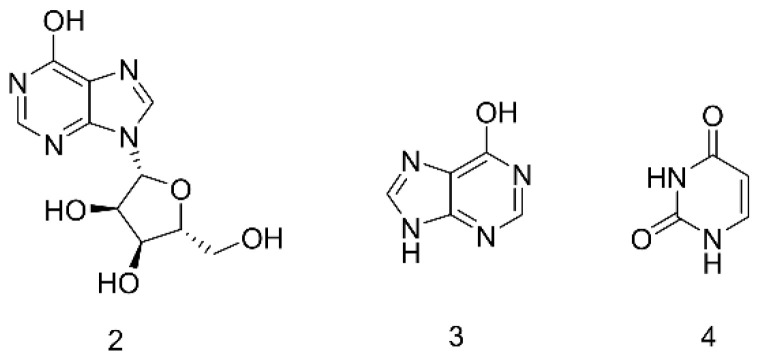
The structure of inosine (**2**), hypoxanthine (**3**), and uracil (**4**) [46].

**Figure 4 molecules-27-08882-f004:**
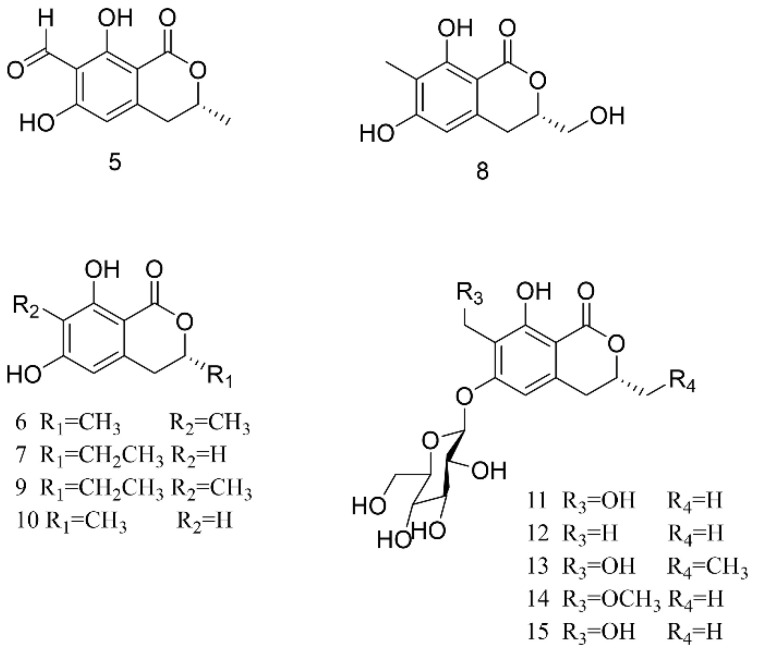
Isocoumarins isolated from *PA* ethanol extract [49,50,51].

**Figure 5 molecules-27-08882-f005:**
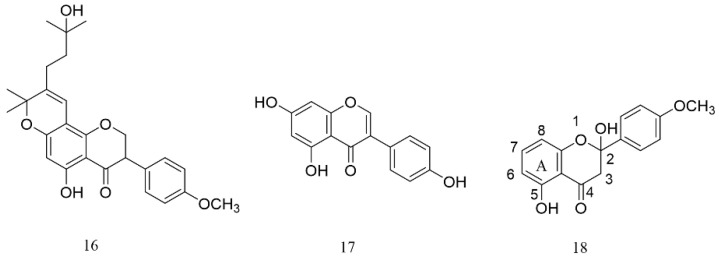
Flavonoids isolated from *PA* ethanol extract [57,58].

**Figure 6 molecules-27-08882-f006:**
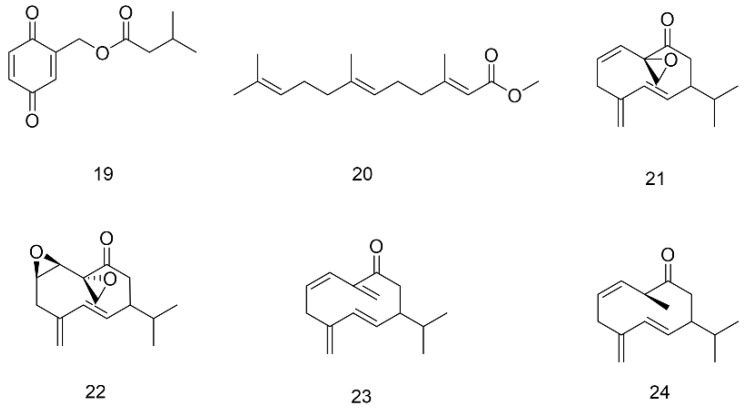
Terpenes isolated from the family Blattidae [64,65,66].

**Figure 7 molecules-27-08882-f007:**
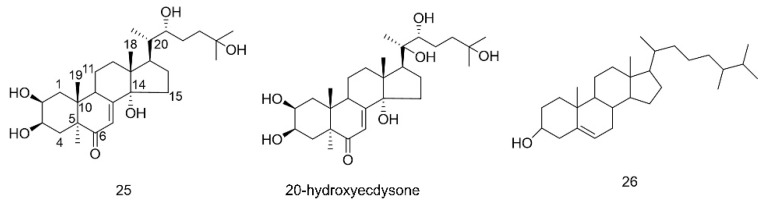
Steroids isolated from *PA* [57,67,68].

**Figure 8 molecules-27-08882-f008:**
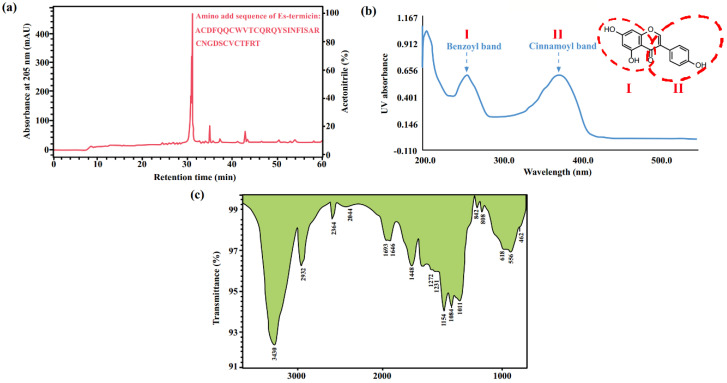
(**a**) The fraction Es-termicin indicated purified by RP-HPLC C_18_ column, (**b**) UV spectrum of flavonoids, (**c**) IR spectrum of polysaccharides.

**Figure 9 molecules-27-08882-f009:**
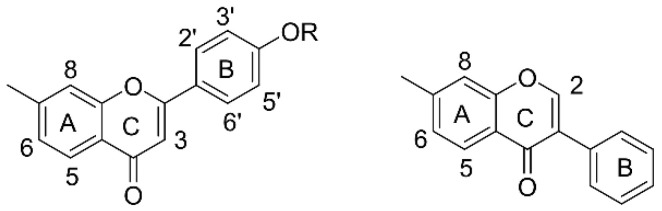
Structures of Flavonoids.

**Figure 10 molecules-27-08882-f010:**
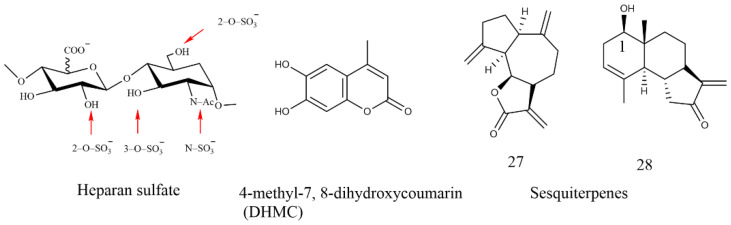
Structures of Heparan sulfate, DHMC, sesquiterpenes [108,109,110,111,112].

**Figure 11 molecules-27-08882-f011:**
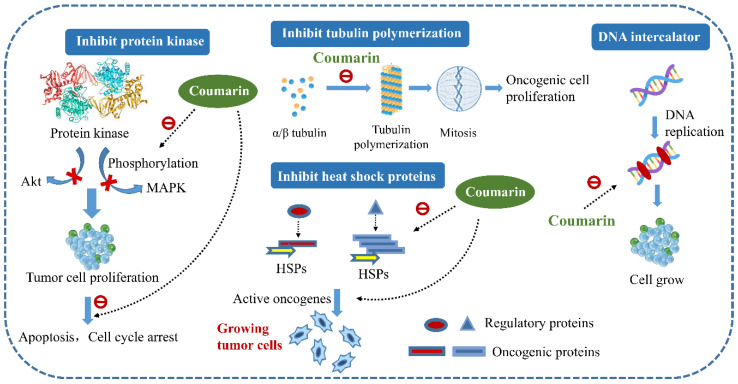
The antitumor mechanism of coumarin.

**Table 1 molecules-27-08882-t001:** Composition and mass fraction of AAs in *PA* [20,21].

Name of AAs	Mass Fraction/%	Name of AAs	Mass Fraction/%
Aspartic acid #	3.67	Phenylalanine *#	1.73
Threonine *	1.82	Lysine *#	2.59
Serine	1.80	Histidine	1.16
Glutamic acid #	5.08	Arginine #	2.26
Glycine #	2.84	Proline	2.64
Alanine	3.35	T	43.14
Cystine	2.55	E	15.26
Valine *	3.38	N	27.88
Methionine *#	0.71	F	8.75
Isoleucine *#	1.84	E/N	54.73
Leucine *#	3.19	E/T	35.37
Tyrosine	2.53	F/T	20.28

*: Essential AAs; #: medicinal AAs; T: total AAs; E: essential AAs; N: eon-essential AAs; F: flavor AAs = aspartic acid + glutamic acid.

**Table 2 molecules-27-08882-t002:** Examples of extraction and separation methods for the family Blattidae.

Methods	Solvents/Material	Conditions	Target Components	Pros and Cons	References
Percolation	90% ethanol	Defatted powders; 10 equivalents of 90% ethanol at 1 mL/20 s by three times	Peptides		[73]
Refluxing	75% ethanol	Dry powders; 15 equivalents of 75% ethanol for 1 h by two times	Active components with antitumor activity		[73]
Ultrasonic assistance	90% ethanol	Defatted powders; 10 equivalents of 90% ethanol; sonicated for 20 min by three times (power: 240 W)	Peptides and amino acids		[73]
Salting-out	Ammonium sulfate solution	Staged salting out with different saturation of 0~35%, 35~70%, 70~100%	Proteins		[79]
Ultrafiltration	Hollow fiber ultrafiltration membrane with a molecular weight cutoff of 3 kDa	Speed:4500 r/min, concentration ratio: 30%; pH:11; 30 min; dialyzed with Tris-HCl buffer solution (0.01 mol/L pH 7.60)	Small molecular peptides	Simple structure, convenient operation, low energy consumption	[81]
Conventional Column Chromatography	HP20 resin	2 cm × 30 cm, 30 g resin; 0.3 g/mL of extract; 2 BV/h, 2 BV; eluted with 2 BV of 70% ethanol at 2 BV/h	Peptides and amino acids	Large adsorption capacity, fast adsorption speed, high elution rate, and can be regenerated and used	[83]
Conventional Column Chromatography	Silica gel	Chloroform extraction site; gradient elution with V_cyclohexane_:V_ethyl acetate_ = 100:0~0:100, V_ethyl acetate_:V_methanol_ = 100:0~50:50.	Phenylpropionic acid, phenylacetic acid, esters, quinolinone		[53]
Conventional Column Chromatography	Silica gel	The ethyl acetate extract (400 g); gradient elution with petroleum ether, petroleum ether/ethyl acetate (15: 1, 8:1, 5:1, 3:1, 1:1), ethyl acetate, ethanol	Oleic acid-1-glyceride and linoleic acid-1-glyceride		[84]
Conventional Column Chromatography	Silica gel	silica gel (200–300 mesh); gradient elution with V_chloroform_:V_methanol_:V_water_ = 9:1:0.1~0:0:1	3,6-dimethyl -2,5-piperazinedione and L-hydroxyproline		[85]
Preparative high-performance liquid chromatography	Spherisorb C_18_ column	Flow rate: 1 mL/min; mobile phase: acetonitrile in water containing 0.1% trifluoroacetic acid (TFA); detector wavelength: 230 nm	Two proteins of 60 and 72 kDa	High separation efficiency and wide application range	[28]

**Table 3 molecules-27-08882-t003:** Examples of pharmacological activities of the family Blattidae.

Activities	Active Substances	Target Object	Mechanism	References
Antitumor	Kangfuxin Liquid (IC50 = 13.99 mg/mL)	BGC-823	To inhibit the proliferation of tumor cells and reduce the number of S-phase cells	[124]
Antitumor	CII-3 (Extract of *PA*) combined with cisplatin (100 mg/kg suspension of CII-3)	Lewis lung cancer model in mice	To prevent angiogenesis near tumor cells	[125]
Antitumor	Extract of *Eupolyphaga sinensis* Walker (IC50 = 0.13 mg/mL)	Human Hepatoma Cells SMMC-7721	To prevent angiogenesis near tumor cells	[126]
Antifibrosis	Water extract of *PA* (200 mg/kg)	acute immunological liver injury caused by concanavalin A (ConA) in mice	To reduce the level of MDA; increase that of SOD and GSH	[127]
Antifibrosis	Glycosaminoglycans (120 mg/kg)	Chronic alcoholic hepatic injury in rats	To reduce the level of inflammatory factors in the body and prevent lipid peroxidation	[128]
Wound-healing	W_11_-a_12_(Extract of *PA*, 10 mg/wound)	Neutrophils	To improve the spontaneous and chemotactic functions of neutrophils, enter the wound for phagocytosis, remove necrotic tissue	[47]
Wound-healing	Kangfuxin liquid (200 μL, 5.3 mg/mL)	Granulation tissue	To promote the proliferation of granulation tissue and mediate the mucosal repair effect of epithelial cells	[90]
Anti-inflammatory	CII-3 (Extract of *PA*, 200 mg/kg)	Swelling of auricle model induced by dimethyl benzene	To reduce the content of PGE2, histamine, and MDA in the inflammatory part and increase SOD activity	[120]
Antibacterial	The antibacterial peptides (20 μL, 1.21 mg/mL)	Gram-negative and positive bacteria	To dent and perforate the outer wall of the bacteria, the substance in the bacteria leaks, and then the bacteria disintegrate	[28]
Myocardial protection	Xinmailong injection (60 mg/kg)	Cardiovascular system	To promote Ca^2+^ inflow of myocardial cells and lastingly increase the positive muscle strength of the heart	[122]
Antioxidant	Polysaccharides (IC50 = 0.311 mg/mL)	hydroxyl free radicals (·OH)	To scavenge hydroxyl radicals	[123]

## Data Availability

All reported or analyzed data in this review are extracted from published articles.

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
