# Peer review of "Phytochemical Profiling, Isolation, and Pharmacological Applications of Bioactive Compounds from Insects of the Family Blattidae Together with Related Drug Development"

_molecules, 2022, doi:10.3390/molecules27248882_

Round 1
Reviewer 1 Report
1.The topic of this article is the medicinal Blattidae, so I suggest that the description of vector mosquitoes and insecticides in the introduction can be reduced. Focus on the application and development of Blattidae
2.Is Table 1 Figure 2-7 from the author's research data in this paper or is it cited from other literature?
I think this is an interesting article about the use of functional components of Blattidae. If possible, can you supplement some information on the successful use of Blattidae as health care products/drugs in the market?
3.table 3, it is recommended to write the dose of the compound
Reviewer 2 Report
This review by Jing Li and Shun Yao focuses on analyzing the current studies and utilization of medicinal insects in the family Blattidae. This work is expected to provide meaningful and valuable information and promote further exploration and development of lead compounds or bioactive fractions for new drugs from the insects.
Some mistakes were found:
1) in Figure 3. The structure of inosine (2), hypoxanthine (3), and uracil (4): Stereochemistry of the inosine (2) is not clear.
2) Page 10, periplanone B(22) is not correct (sesquiterpenes[68]), and structure 24 is deformed.
3) Page 10, The structure of ecdysone 25 is not correct (Me –> H)
3) Page 22, The structure of sesquiterpene 28 is not correct (oxygen of lactone ring is missing).
